# Post Fire Residual Strength of the Wall-Slab Using Siliceous Concrete

**DOI:** 10.3390/ma14071793

**Published:** 2021-04-05

**Authors:** Su-Hyeon Lee, Byong-Jeong Choi

**Affiliations:** Department of Architectural Engineering, Kyonggi University, Suwon 16227, Korea; hyeon1284@naver.com

**Keywords:** siliceous concrete, wall-slab connection, residual strength, reduction factor, full-scale fire test, simple calculation method (SCM)

## Abstract

It is very important to understand the residual performance of a structure for repair, retrofit, and reuse of a building after a fire. In this study, an experiment is conducted on the residual performance of real-scale siliceous aggregates-based reinforced concrete (RC) wall-slab connection (WSC) after the fire, using the simple calculation method (SCM) of standards (Eurocode, ACI, and NIST) for comparison and analysis. A description of the WSC specimen and detailed methods for the experiment are introduced. The fire test is conducted according to the fire scenario by dividing it into one-sided and two-sided heating based on the wall. In the post-fire residual performance test, the load–displacement and moment-deflection angle relationship according to the fire time are derived and discussed. In addition, the residual mechanical properties after the fire are derived for the 35 MPa siliceous concrete used in the wall-slab specimen. The load and moment, derived using SCM, are compared with the experimental results. Our results show that the one-sided heating test result is close to that of Eurocode’s SCM, and the two-sided heating test result is close to that of ACI (NIST)’s SCM. This study provides a database on the residual strength through a real-scale fire test and standard comparison.

## 1. Introduction

If the strength and stiffness of reinforced concrete (RC) structures decrease, due to long-term fire damage, the entire RC building may collapse [1]. From a structural point of view, buildings with fire damage should be repaired or reinforced, or in serious cases, replaced [2]. Recently, many large-scale fires in buildings have occurred in Korea, and residual performance evaluation is conducted to ensure the structural safety of buildings in which fire damage occurred. If we easily catch up to the residual capacity or strength of the damaged structure after the fire, it will definitely benefit all structural fire engineers.

There are some research works have been carried out to evaluate the residual strength of a structure using various methodologies after the fire. Choi, K.H [3] used non-destructive rebound-hardness and ultrasonic testing methods to measure the residual strength of reinforced concrete structures. Those techniques were also applied to assess the stiffness and deflection of RC slabs members. But the measurements were not developed further in structural member connections. Also, it is not easy to derive the mechanical performance of the entire structure from the local non-destructive test (NDT) data extracted at the structural member. Khoury [4] focused on the effect on state-of-the-art concrete and concrete structures. Khoury mentioned the deterioration of concrete in mechanical properties, the influence of transient creep, influence of loading during heating, and effect of temperature upon the residual compressive strength of high-performance concretes after heat cycling. The paper does not fully represent the residual strength of the post-fire structural member connection scenarios. However, his study became a basis for performance-based fire engineering by suggesting lots of testing datum of high strength concrete [5,6].

Kodur et al. [7] suggested a numerical model for predicting the fire resistance of reinforced concrete beams. Kodur et al. knew the current prescriptive approaches for evaluating fire resistance do not represent the realistic fire, loading, and fire scenario in practice. Thus, he focused on finding the type of failure criterion of fire resistance of RC beams using many numerical models. Their methodology also progressed into the complex member connection of RC structures using numerical models. 

Guo et al. [8] performed experimental works to acquire the structural behavior of composite slab of seven specimens. Their tests have included variables, such as fire scenarios to develop the temperatures, vertical deflections, horizontal deflections, support reactions, and deck strains of the composite slabs. But the design equations under the fire were not suggested and applied to structural member connections. 

Li et al. [9] worked on the experimental studies on seismic performance of post-fire of reinforced concrete frames. In their cyclic loading tests, the failure mode, ultimate strength, vertical, and lateral displacements, ductility, and initial stiffness were investigated. However, the design strengths using Eurocode or ACI fire equations were not compared reciprocally.

Li and Guo [10] investigated the heating and cooling effects on the restrained steel beam. Even though they used simple steel beams, their tests endeavored to have restrained beams by connecting them to columns in both ends. The axial forces of the restrained beams were greatly affected by the cooling effects. 

Wang et al. [11] carried out the performance tests by the modeling of reinforced concrete slabs in the fire. In their studies, temperature distributions of the RC slab specimens were well estimated in terms of the critical temperature ranges. The stress-strain relationship, reduction factor, peak strain, free thermal strain, transient strain, and creep strains were validated with moisture contents.

Kodur et al. [12] evaluated the flexural capacity of RC beams after exposure to fire for both normal-strength concrete and high-strength concrete. The variation of residual strength of reinforcing steel and concrete as a function of temperature was well evaluated to be applied to the residual flexural strength of the beam, as shown in Equation (1).
(1)$$M_{n}=A_{s}f_{yT}\left(d^{*}-A_{s}f_{yT}/1.7b^{*}f_{c}^{\prime }\right)$$

This equation means that if the residual strength of the steel reinforcement is evaluated, the flexural capacity of the beam after the fire can be computed using an ambient temperature strength design equation mentioned in codes (e.g., ACI 318 (ACI, 2008), Eurocode 2 (ECS, 2004). 

Choi et al. [13] predicts the temperature of the hollow slab using the Wickström’s Method and the proposed temperature calculation equation, and compares the moment and deflection with the experimental results. The simple calculation method Equation (2) and the fire damage cross-sectional moment Equation (3) presented in Eurocode 2 were used. The material (concrete, reinforced bar) strength reduction factor was calculated using the strength reduction factor calculation method of ACI 216 and Eurocode 2. When compared with the experimental results, only 2.7% error occurred from the proposed method, so a reliable result was derived. However, due to the characteristics of the hollow slab, additional research was required in consideration of the shape, size, and hollow location.
(2)$$M_{f}=A_{s}f_{y,T}\left(d-a_{f}/2\right)$$
(3)$$M_{n,\theta }=\left(A_{s}-A_{s}^{\prime }\right)f_{y,\theta }\left(d-a_{\theta }/2\right)+A_{s}^{\prime }f_{y,\theta }\left(d-d^{\prime }\right)$$

Liao et al. [14] modeled local fracture of RC beams at high temperature using XFEM (extended finite element analysis) nonlinear method. The XFEM nonlinear analysis procedure for a simply supported RC beam showed high accuracy when compared with the previous fire test results. 

Vecchio et al. [15] made 12 new Toronto beams by referring to 12 beams of Bresler–Scordelis (1963) previously studied. There were slight differences in the material properties and structural details of the new Toronto beam, but it was reproduced with high accuracy. Vecchio provided higher quality data compared to previous studies.

Sucharda et al. [16] performed 3D nonlinear finite element analysis using test data from Vecchio [15]. The accuracy of the simulation results based on the measured compressive strength was satisfactory. However, the actual design noted that it was important to properly consider the random characteristics of the input data in consideration of the required level of safety and service.

Until now, we investigated various research works to grasp the integrity of residual strength for the various structural members. However, the current methods for post-fire assessment were still based on a lack of scientific basis and needed various accumulation of fire data covering the residual strength evaluation of structural member connections. Therefore, this paper tried to establish the design methodology and assessment of the residual strength using the RC slabs connected to the wall with the post-fire scenario. The slab to wall connection is a commonly used structures in practice where the evaluation of the residual strength is affected by the deflection angle and reduction factor of both concrete and reinforcement materials. The deflection angle will be affected by the applied moment, due to the fire placed in one side or two sides at the same time. Recently, the scope of research has been extended to structural performance in the cooling stage after fire damage, and many numerical and finite element analysis (FEA) techniques have been developed to predict the fire behavior of real-scaled structures [1,7,11,12,13]. In this study, post-fire experiments were conducted to find out the residual performance of the RC slab specimens. It is also very crucial to assess the residual strength performance for the complex structural members, such as slab to wall connection. 

The type of the collapse condition is can be differed, as shown in Figure 1a,b, where the results of residual strength could be different from each other. The two supporting ends of the RC slab can be divided into simply supported, shown in Figure 1a, and continuously supported, shown in Figure 1b. In the case of a simply supported RC slab, the tensile strength of the concrete is ignored, and only the reduction in the yield strength ($f_{y,T}$) of the reinforcement is considered, due to the heated concrete at the positive moment at the bottom. Because it is also conservative to ignore compression hardening in the compression region, the calculation is relatively simple. In the case of a continuously supported RC slab, the load is transferred to another location before the collapse mechanism occurs using the moment redistribution. As shown in Figure 1b, a three-hinge is required to collapse in the continuously supported RC slab where the experiments are too complicated to understand.

Because not only the simply supported RC slab has sufficient to evaluate the fire resistance, but additional calculation at the support is convenient to understand after experiments. The paper chose simply supported end conditions, rather than continuous supports. The simply supported condition is enough to evaluate the residual strength after the fire for the RC slab specimen, which is also a conservative with the aggravated condition. In this study, moment results and experimental results were compared using simple calculation methods (SCM) of Eurocode [17], ACI [18], and NIST [19] for simply supported slabs. The testing set-ups will be introduced in the next chapter in detail with further explanations. By comparing various codes and real-scale experiments, it is possible to secure expanded data compared to existing studies. Now the main purpose of the paper is to investigate the post-fire residual strength by applying the material reduction factors of the slab to the wall connection specimen. To find out the residual strength, the moment and deflection angle were evaluated carefully under the vertical loading after the cooling of the damaged specimen by the fire scenario conditions. Secondly, the paper endeavored to examine the post-fire flexural residual strengths and moments given the current standard using EN1992-1-2 [17], ACI 216 [18], and NIST1681 [19] comments. Thirdly, the paper examined the damage effects of the two types of fire scenario with one side fire and two-side fire conditions. The post-fire damage effects of the specimens subjected to vertical loading are compared with the variations, such as fired heating times, load–displacements, reduction factors, moment-deflection angles, and validation of the current standards with the two fire scenarios. These experimental works could be useful to understand the current fire design equation and their post-fire performance under real fire conditions. The tested material, set-ups, fire scenarios, and experimental methods will be discussed in the next chapters.

## 2. Wall–Slab Connection Residual Performance Test

In this chapter, the residual mechanical properties of materials (35 MPa siliceous concrete, reinforcing bar), the manufacture of the specimen, and the test plan were presented and discussed to derive the residual strength of the wall-slab connection. The residual mechanical properties of concrete and reinforcing bar (rebar) after the fire are entered in simple calculation methods (SCM) of EN1992-1-2 [17], ACI 216 [18], and NIST1681 [19]. Detailed test specimen manufacture and planning can increase the reliability of data and can be used as basic data for further numerical and finite element analysis studies.

### 2.1. Material Test

Currently, the standard does not present the residual strength of reinforcing bars and concrete for cooling conditions after the fire. After cooling of rebar or concrete that has been damaged by fire, the mechanical properties of the material change. Steel materials, such as rebars that are exposed to high temperature, are known to increase or maintain their strength after cooling below about 600 °C [1,20,21]. On the other hand, it is known through many studies that the strength of concrete exposed to high temperature is not recovered, and after cooling is decreased [22,23,24]. Therefore, in this study, a cooling test after heating was performed to determine the residual performance of the concrete (${f^{\prime }}_{c}$ = 35 MPa) used in the WSC specimen. In general, the cooling test after heating can be divided into a ‘cooling in air test’ (CIA), and a ‘cooling in water test’ (CIW). The residual strength of the concrete specimen was derived under the pre-unloaded (without applying force to the specimen) CIA condition, which is the same environment as the wall–slab specimen. Figure 2 shows that the CIA test for the concrete specimen was divided into four stages. First, the concrete specimen is heated at 5 °C/min in a furnace to the target temperature without applying force (step 1). After confirming that the thermocouple inside the concrete specimen has reached the target temperature, the temperature is maintained at a steady state at the target temperature for at least 30 min (step 2). After that, the heat source is removed from the concrete specimen, and the specimen is gradually cooled in the electric furnace (step 3). The specimen that has cooled sufficiently in the electric furnace is removed, and is sufficiently cooled at room temperature (RT; 20 °C) for about 2–3 days. Then, a compression test is performed on the cooled concrete (step 4). For the cooling effect test data of concrete, the average value of the results obtained by performing the test in triplicate for each target temperature is used.

A thermocouple was installed inside the concrete specimen, and after 28 days of underwater curing, air–dry curing was conducted in a constant temperature and humidity room (20 °C, 60% R.H.). Heating conditions were raised at a temperature ramp of 5 °C/min until reaching the target temperature, which was then maintained at a steady state for 30 min or longer, and cooled by removing the heat source (Figure 2). The compression test was conducted according to ASTM (American Society for Testing and Materials) C 39 [25]. The test on the residual mechanical properties of the siliceous aggregates concrete cylinder specimens were carried out three times for each temperature. The residual compressive strength ($f_{c,\theta }$), residual elastic modulus ($E_{c,\theta }$), average value of residual compressive strength (${f^{\prime }}_{c,\theta }$), average value of residual elastic modulus (${E^{\prime }}_{c,\theta }$), standard deviation, and reduction factor of siliceous aggregates concrete are summarized in Table 1. Table 1 shows that the average compressive strength (${f^{\prime }}_{c,20}$) and elastic modulus ($E_{c,20}$) of concrete at RT were derived as (41.93 and 23,700) MPa, respectively. As a result of the concrete compression test after cooling, the strength of the concrete cooled from 100 °C decreased to 38 MPa, which was 91% of the RT compressive strength. The elastic modulus at 100 °C was 20,633 MPa, which fell to the level of 87% of the RT elastic modulus. The strength of the concrete cooled at 200 °C was 35.30 MPa, which was 84% of the compressive strength at RT. The elastic modulus decreased to 76% of the RT elastic modulus at 18,033 MPa. Afterwards, the compressive strength and elastic modulus of concrete specimen continuously decreased in all temperature ranges from (300 to 900) °C. The strength of the concrete cooled at 900 °C was 3.8 MPa, which was 9% of the compressive strength at RT. The elastic modulus at 900 °C was 433 MPa, which fell to the level of 2% of the RT elastic modulus. The strength of the concrete cooled at 1000 °C was 2.63 MPa, which was 6% of the compressive strength at RT. The elastic modulus at 1000 °C was 233 MPa, which fell to the level of 1% of RT elastic modulus. At 900, 1000 °C, the compressive strength rapidly decreased, due to the spalling effect. At high temperatures, concrete causes dehydration of cement paste and aggregate, and cracks occur, due to the difference in the thermal expansion rate. In addition, the residual strength of the cooled concrete decreases further, due to microcracks [1,23,26]. Through cooling tests, it was confirmed that the concrete exposed to high temperature had lower strength after cooling. Kee et al. [27] presented other residual mechanical properties (inner temperature, poisson’s ratio, dynamic, and static elastic modulus, etc.) for 35 MPa siliceous concrete using the non-destructive testing (NDT) method.

Steel materials, such as rebar, are known to be homogeneous compared to concrete, and steel strength does not decrease significantly, even after cooling after exposure to high temperatures [1,20,21]. Figure 3 summarizes the data on the residual yield strength of steel and rebar of previous reports [21,28,29,30,31,32,33], where the yield strength of the steel and rebar study data is mild steel between (300 and 460) MPa, and only the CIA test results were compared. The original strength was generally maintained from RT to 600 °C, and the strength began to decrease from 700 °C. The average value was derived by arranging the existing research result data, as shown in Table 2. As a result of analysis through Figure 3 and Table 2, steel materials, such as steel and rebar, after cooling at high temperature, showed a similar tendency. The residual yield strength of SD400 ($f_{y,20}$ = 400 MPa) rebar used in this WSC specimen was used as the average value of data in the literature. The residual elastic modulus of rebars did not differ by temperature. Therefore, the residual elastic modulus ($E_{y,\theta }/E_{y,20}$) of 1.0 was applied at all temperatures, as shown in Table 2. Where $E_{y,\theta }$ is the elastic modulus of rebar at elevated temperature (MPa), $E_{y,20}$ is the elastic modulus of rebar at 20 °C (MPa).

The residual strength reduction factors of concrete and steel presented in Table 1 and Table 2 were entered into the SCM of Eurocode [17], ACI [18], and NIST [19], to derive moment results. The moment value derived from the residual strength reduction factor was compared with the experimental moment result.

### 2.2. Wall-Slab Connection Specimen Fabrication Plan

In order to determine the residual strength of the WSC after a fire, a test specimen was planned according to the fire occurrence scenario. In an actual building fire, the members are connected to each other, so additional load and axial restraint effects exist, as shown in Figure 4. To bring about live load are used a water tank, sandbag, and actuator in a general fire test. However, fire tests or residual performance tests of some connecting elements having sufficient fire resistance, such as wall-slabs in this study, do not necessarily need to be evaluated by the existing fire resistance test methods [34]. The focus of this study is to derive the residual mechanical properties of siliceous aggregates concrete, and to provide data on the temperature and residual performance of full-scale wall-slabs in various fire situations. Therefore, this residual performance test limited the following items: Live load and other loads (including only self-weight of test specimen);Axial restraint effect;Pressure if the furnace-related to the atmosphere around the specimen;Thermal properties, such as thermal expansion and heat flux.

Figure 4 shows that predictable fire occurrence scenarios include one-sided heating (OS) and two-sided heating (TS), based on the wall. In EN1992-1-2 [17], Table 3 shows that fire resistance grades are presented by categorizing the walls by exposure to fire on one side or two sides. In this study, an experiment was performed assuming the following cases as a one-sided or two-sided heating fire situation: (1) One-sided heating: When the wall is on the outermost part of the structure, or when there is fire on only one side; and (2) two-sided heating: When the wall is inside the structure, and a fire occurs on both sides of the wall. Considering the equipment specifications of the horizontal heating furnace, it was planned in the form of a cross (+), as shown in Table 4, so that the wall and slab could be heated on one or two sides.

Eurocode 1992-1-2 [17] and ACI 216-1M-14 [18] specify the minimum member thickness and minimum cover thickness according to the fire rating of walls and slabs, respectively. First, Table 3 shows that in EN1992-1-2 [17], when the wall is exposed on one or two sides, the minimum member thickness and the minimum cover thickness of the RC structure bearing wall are specified separately. This WSC specimen was designed based on a fire situation for 3 h. Table 4 shows that this test specimen has a wall thickness of 1220 mm and a covering thickness of 50 mm or more, so the fire resistance rating REI 180 (3 h) was satisfied for row 2 (one-sided heating) and row 3 (two-sided heating) of Table 3. Where $u_{fi}$ is the reduction factor for the design load level in the fire situation, as shown in Equation (4), and represents the ratio of the axial design strength ($N_{Rd}$) at RT to the axial load ($N_{Ed,fi}$) in a fire situation. The thickness of the wall of this specimen is 1220 mm and the length is 1500 mm, and since the cross-section is sufficiently large, $u_{fi}$ = 0.35 was assumed for Table 3.
(4)$$u_{fi}=N_{Ed,fi}/N_{Rd}$$

Table 5 shows the minimum member thickness and cover thickness of the slab according to the fire grade of EN1992-1-2 [17]; and since the thickness of the slab of this specimen is 460 mm, the fire resistance grade REI 180 (3 h) or more was satisfied. In addition, the ratio of the long side ($l_{y}$ = 1650 mm) to the short side ($l_{x}$ = 1500 mm) of one arm of the slab is 1.1, as shown in Equation (5), which is less than 1.5, so it is a two-way slab. Therefore, the slab cover thickness of this WSC specimen according to the fire grade EN1992-1-2 [17] corresponds to row 4 of Table 5, and the minimum cover thickness is 30 mm in fire grade REI 180, so the slab cover thickness of this test sample of 40 mm or more was satisfied.
(5)$$l_{y}/l_{x}=1650\;mm/1500\;mm=1.1\;\left(l_{y}/l_{x}\leq 1.5\right)$$

Table 6 shows the minimum member thickness of walls and slabs according to the fire grade of ACI 216.1M-14 [18]. Since the type of concrete aggregate is siliceous, this WSC specimen also satisfies the fire resistance level of 3 h (minimum thickness of 155 mm or more). In addition, Table 7 shows the minimum cover thickness of the wall and slab according to fire grade ACI 216.1M-14 [18]. Since this WSC specimen was in an unrestrained state and a non-prestressed state, the fire resistance grade 3 h criterion (minimum thickness of 30 mm or more) was also satisfied.

### 2.3. Test Method

Table 8 shows that the WSC specimens were made of one non-heating specimen (NS) and six heating specimens, according to the heating time and heating surface. For the heating specimen, three one-sided heating (OS) tests and three two-sided heating (TS) tests were conducted. The heating specimen was heated for (1, 2, and 3) h according to the fire scenario. First, Figure 5a shows that the fire test was conducted using a horizontal heating furnace of the Korea Ship and Offshore Research Institute (KOSORI, in Korea) at Pusan National University, and the heating was performed according to ASTM E119-15 [35]. Figure 6 shows that ASTM E119 [32] was used for the standardized fire-curve, and each was heated for (1, 2, and 3) h, depending on the fire conditions. After heating, the WSC specimen was sufficiently cooled at RT for about (2 to 3) days or more. After heating, the WSC specimen was sufficiently cooled at RT for about (2 to 3) days or more. In order to check the temperature distribution of the WSC in the case of fire, Type-K thermocouples were installed at regular intervals from the wall and slab concrete surfaces, as shown in Figure 7. A total of 22 thermocouples per specimen were used. The maximum operating temperature for Type K thermocouples is 1260 °C. The thermocouple of the wall was located (30, 30, 30, 50, 50, 50, 50, and 100) mm from the fire-exposed surface. The slab thermocouple was located (30, 30, 30, 50, 50, 50, 50, and 100) mm from the fire-exposed surface. Thermocouple T/C Nos. 17 and 18 were located at the wall–slab connection, and the temperature of the connection was measured at about 50 mm from the wall surface. T/C No. 19 was located at 30 mm from the wall surface, and Zone 2 is a non-heated surface in OS, but it was installed to check the temperature of the non-heating surface. T/C Nos. 20, 21, and 22 were installed by depth on the upper rebar of the wall. Therefore, the thermocouple of each wall and slab consisted of a total of 16 concrete thermocouples and six internal rebar thermocouples. The one-sided heating (OS) test was heated only in Zone 1, while the two-sided heating (TS) test was simultaneously heated in Zones 1 and 2.

By installing a thermocouple, it was possible to check the temperature inside the specimen, due to fire, and the strength reduction factors of concrete and rebar according to the temperature could be derived from Table 1 and Table 2. The post-fire loading test was performed using the actuator of the POSCO RIST (in Korea) test building, as shown in Figure 5b, and the residual strength was determined by applying a vertical load to the WSC, until it was destroyed. A total of 6 LVDTs (Linear Variable Displacement Transducer) were installed, as shown in Figure 8. Vertical displacement was measured by installing two horizontal LVDTs (LVDT 1, LVDT 2) to measure the slab end, and two vertical LVDTs (LVDT 3, LVDT 4) at 100 mm intervals on both sides of the wall. In addition, two vertical LVDTs (LVDT 5, LVDT 6) were installed in front of and behind the central part of the wall, to measure the vertical displacement of the central part. The position of the support stage was installed at the center of each slab.

## 3. Fire Test Results and Discussion

### 3.1. Temperature Distribution

For one-sided (OS) or two-sided (TS) heating specimens of WSC, Table 9 summarizes the temperature and maximum temperature of OS and TS heating thermocouples by depth from the fire-exposed surface according to the fire time. Figure 7 summarizes the temperature distribution of the specimen according to the thermocouple position. At the same thermocouple location, a temperature difference occurred according to the fire time. The maximum temperature of the OS of the slab thermocouple T/C No. 1, located closest to the fire surface, rose to (250, 333, and 1061) °C for (1, 2, and 3) h, respectively. The reason that the temperature at the bottom of the slab increased sharply at 3 h seems to be that the thermocouple was exposed to the fire, as the concrete fell out, due to the spalling effect, as shown in Figure 9a. The maximum temperature of TS heating T/C No. 1 rose to (34, 89, and 124) °C for (1, 2, and 3) h, respectively. The reason that the temperature difference between OS and TS occurred in thermocouple T/C No. 1 is that in the TS test specimen, only the spalling effect on the wall occurred, and not the slab, as shown in Figure 9b. The maximum temperature of the OS test of the wall thermocouple T/C No. 8, located closest to the fire-exposed surface, rose to (152, 128, and 218) °C for (1, 2, and 3) h, respectively. The maximum temperature for the TS test was (208, 216, and 389) °C for (1, 2, and 3) h, respectively. The maximum temperature of the OS specimen of the slab bottom rebar thermocouple T/C No. 15 rose to (143, 245, and 866) °C for (1, 2, and 3) h, respectively.

In the 3 h heating test, the reason that the thermocouple temperature at the bottom of the rebar of OS heating increased sharply is that the rebar was exposed to fire, due to the dropping of concrete because of the spalling effect, as shown in Figure 9. The maximum temperature of the OS specimen of the thermocouple T/C No. 16 of the upper rebar of the slab was (26, 24, and 45) °C for (1, 2, and 3) h, respectively. The maximum temperature of the TS specimen was (28, 28, and 35) °C for (1, 2, and 3) h, respectively. In contrast to the lower rebars that were directly exposed to fire, the upper rebars suffered no effect from the fire, due to the thickness of the cover. The maximum temperature of the OS specimen of the connection thermocouple T/C No. 17 in Zone 1 rose to (318, 445, and 481) °C for (1, 2, and 3) h, respectively. The maximum temperature of the TS specimen rose to (318, 445, and 481) °C for (1, 2, and 3) h, respectively. The OS temperature of the connection part thermocouple T/C No. 18 located in Zone 2 based on the OS specimen was 26 °C for all of the (1 to 3) h cases. Since Zone 2 is located on the opposite side of the Zone 1 heating surface, there was no fire effect. T/C Nos. 17 (Zone 1) and 18 (Zone 2) based on TS specimen are symmetrical based on the wall, as shown in Figure 7, and were under the same heating conditions. Therefore, for conservative evaluation, the temperatures of Zones 1 and 2 were compared, and the maximum temperature was considered as the fire temperature of Zones 1 and 2. The maximum temperature of T/C Nos. 20–22 at the top of the wall was generally maintained at (25–30) °C, so there was no significant effect from fire. 

In EN1992-1-2 [17], the temperature can be predicted according to the depth (X) and fire resistance level from the fire-exposed surface of the slab, as shown in Figure 10. If the temperature is predicted according to EN1992-1-2, the temperature according to each fire resistance grade at a depth of 40 mm of the slab fire-exposure surface is (300, 460, and 560) °C at R60, R120, and R180 (points A, B, and C), respectively. In ACI 216.1M-14 [18], it is possible to predict the temperature according to the type of aggregate and the depth of the slab fire-exposure surface, as shown in Figure 11. The temperature predicted according to ACI 216.1M-14 is (320, 480, and 600) °C at (60, 120, and 180) min, at points A, B, and C, respectively. Table 10 compares the temperature of the thermocouple by main depth (30, 40, 50 mm) according to the fire test results, and the temperature according to EN1992-1-2 [17] and ACI216.1M-14 [18] standards. Data is displayed in Table 10 as columns (A to I) and rows (1 to 9). For example, the 1 h test result of T/C No. 1 of concrete slab is 250 °C, and is located at A1. The load and moment were calculated using the reduction factor derived based on the temperature of each major depth, and compared with the experimental values of the residual strength. 

The thermocouple numbers for each major location for comparison with the experimental results are (1, 8, 15, and 17). T/C Nos. 1 and 8 are concrete thermocouples of slab and wall closest to the fire, respectively, and T/C Nos. 15 and 17 are the rebar thermocouples of the slab and wall closest to the fire, respectively. Therefore, each thermocouple is located in the most vulnerable place from the fire, and can be evaluated conservatively through the maximum temperature. The thermocouple numbers were selected as the main locations to obtain the design strength of the slab. Table 10 shows the temperature data for deriving the reduction factor (k) and reference Figure 10 and Figure 11, while Table 1 and Table 2 were used for the yield strength and elastic modulus reduction factor of concrete and rebar. The 1 h test temperature ($T_{test}$) of the thermocouple T/C No. 1 located at 30 mm thickness of the slab is a maximum 250 °C (A1), the corresponding compressive strength reduction factor ($k_{f,test}$) is 0.76 (D1), and the reduction factor of the elastic modulus ($k_{E,test}$) is 0.64 (G1). On the other hand, at the slab cover thickness of 30 mm (X = 30 mm), 1 h (R60) temperature ($T_{EC}$) according to Figure 10 of EN1992-1-2 is 400 °C (B1), the corresponding yield strength reduction factor ($k_{f,EC}$) is 0.62 (E1), and the reduction factor of the elastic modulus ($k_{E,EC}$) is 0.39 (H1). The 1 h temperature ($T_{ACI}$) according to Figure 11 of ACI 216.1M-14 is 420 °C (C1), the corresponding yield strength reduction factor ($k_{f,ACI}$) is 0.60 (F1), and the reduction factor of the elastic modulus ($k_{E,ACI}$) is 0.36 (I1). Compared to the experimental temperature, the temperature of EN1992-1-2 and ACI 216.1M-14 was derived from (150 to 170) °C higher. Therefore, the strength of EN 1992-1-2 [17] and ACI 216.1M-14 [18] decreased, compared to the experimental results. The 2 h test temperature ($T_{test}$) of T/C No. 1 is maximum 333 °C (A4) in Table 10, the corresponding compressive strength reduction factor ($k_{f,test}$) is 0.65 (D4), and the reduction factor of the elastic modulus ($k_{E,test}$) is 0.48 (G4). On the other hand, the 2 h (R120) temperature ($T_{EC}$) of EN1992-1-2 is 560 °C (B4), the corresponding yield strength reduction factor ($k_{f,EC}$) is 0.43 (E4), and the reduction factor of the elastic modulus ($k_{E,EC}$) is 0.14 (H4). The 2 h temperature ($T_{ACI}$) of ACI 216.1M-14 is 600 °C (C4), the corresponding yield strength reduction factor ($k_{f,ACI}$) is 0.38 (F4), and the elastic modulus reduction factor ($k_{E,ACI}$) is 0.08 (I4). As with the results of the 1 h test, the yield strength and elastic modulus decreased more than the test as the temperature was higher than the test temperature at 2 h of EN1992-1-2 and ACI216.1M-14. The 3 h test temperature ($T_{test}$) of T/C No.1 is up to 1061 °C (A7) in Table 10, the corresponding yield strength reduction factor ($k_{f,test}$) is 0 (D7), and the reduction factor of the elastic modulus ($k_{E,test}$) is 0 (G7). On the other hand, the 3 h (R180) temperature ($T_{EC}$) of EN1992-1-2 is 690 °C (B7), the corresponding yield strength reduction factor ($k_{f,EC}$) is 0.29 (E7), and the elastic modulus reduction factor ($k_{E,EC}$) is 0.05 (H7). The 3 h temperature ($T_{ACI}$) of ACI 216.1M-14 is 710 °C (C7), the corresponding yield strength reduction factor ($k_{f,ACI}$) is 0.27 (F7), and the reduction factor of elastic modulus ($k_{E,ACI}$) is 0.05 (I7). Unlike the previous (1 and 2) h results, as the 3 h test temperature was higher than the standard, the yield strength and elastic modulus of the experiment were lowered. In addition, the yield strength reduction factor and the reduction factor of the elastic modulus of the rebar were maintained above 0.97 in the fire test for (1 or 2) h. However, in the 3 h fire test, the rebar yield strength reduction factor of the slab decreased to 0.84 (D8), as shown in Table 10. This seems to be the reason for the temperature rise, due to the exposure of the rebar as the concrete fell off because of the spalling effect in the 3 h specimen. This ‘spalling effect’ phenomenon has also appeared in previous experimental studies and has been discussed [4,6,9,12]. In the experiment of this study, like that of Li et al. [9] and Kodur et al. [12], water vapor was generated 15 to 18 min after the start of the fire test, and many microcracks were generated. At 30 min from the start of the experiment, the temperature continued to rise, and a loud sound was generated, due to the explosion (explosive spalling). These phenomena are known to occur for 20 to 30 min after a fire occurs [4]. The prediction model or mechanism, due to explosion heat, should be considered, but for simplicity, intensity reduction, due to explosion heat phenomenon, was excluded. These phenomena are known to occur for 20 to 30 min after a fire occurs [4]. The prediction model or mechanism, due to spalling, should be considered, but strength reduction, due to spalling, was excluded for simplicity.

### 3.2. Load–Displacement Relationship

The WSC specimen was subjected to a pre-fire test for heating one or two sides, and a sufficient cooling period of at least three days was performed. A monotonic loading test was performed using an actuator on the upper part of the wall of the cooled WSC specimen. Figure 12 shows the load–displacement comparison relationship of the non-heating specimen (NS) and the one-sided heating specimen (OS), where the maximum load (kN, P) is the load of the actuator, and the displacement (mm, δ) is the average value of LVDT 5 and 6, as shown in Figure 8 installed in the lower center of the wall. 

In each relationship, the position of the maximum load for the maximum displacement is marked with a symbol. The non-heating specimen (NS) had a maximum load of 1779.62 kN, and a maximum displacement of 18.11 mm. One-sided heating 1 h specimen (OS-1H) had a maximum load of 1757.56 kN, and a maximum displacement of 12.40 mm. One-sided heating 2 h specimen (OS-2H) had a maximum load of 1926.21 kN, and a maximum displacement of 13.65 mm. The difference in the maximum load between NS and OS-1H was about 22 kN, which was not significantly different, but the maximum load of OS-2H increased by about 150 kN, compared to NS. The temperature of the tension rebar in the positive moment part of OS-2H was 245 °C (A5), as shown in Table 10. According to Table 2, in the case of HB Carbon Steel [31], A572 Gr.50 [32], Q460 [33], and A992 [21], the strength increased after cooling between (200 and 300) °C. Also, the temperature of the rebar at the connection was 445 °C (A6), as shown in Table 10. According to Table 2, S460 (Strain Level 0.2%) [30], A572 Gr.50 [32], A992 [21], and Q460 [33] all increased in strength after cooling in the range (400 to 500) °C. Due to the increase in strength of the tensile rebar in the positive moment part of OS-2H, the strength of the test specimen increased. On the other hand, the one-sided heating 3 h specimen (OS-3H) had a maximum load of 856.83 kN, and a maximum displacement of 10.52 mm. Compared to NS, the strength of OS-3H was about 48%, as shown in Table 11. In OS-3H, as shown in Table 10, spalling effect and dropping of concrete occurred within 3 h, resulting in a sharp decrease in concrete strength to 0% (D7). In addition, the rebar also recovered some strength, but the strength was 84% (D8), compared to the RT strength. Figure 13 shows the load–displacement comparison relationship of the non-heating specimen (NS) and the two-sided heating specimen (TS). The two-sided heating 1 h specimen (TS-1H) had a maximum load of 1043.66 kN, and a maximum displacement of 9.99 mm. Compared to NS, the strength of TS-1H decreased to 59%, as shown in Table 11. The two-sided heating 2 h specimen (TS-2H) had a maximum load of 901.98 kN, and a maximum displacement of 15.39 mm. The load ratio of TS-2H to NS was 51%, as shown in Table 11. The two-sided heating 3 h specimen (TS-3H) had a maximum load of 856.34 kN, and a maximum displacement of 14.16 mm. The strength of TS-3H (856.34 kN) was the same as that of OS-3H (856.83 kN), and it was 48% of the maximum load of NS. Figure 14 summarizes the load–displacement relationships of the NS, OS, and TS results, while Table 11 summarizes the loads and displacements of each specimen. The load ratio ($P/P_{NS}$) to the maximum load (P) of the NS specimen was derived.

Table 11 and Figure 14 show that the strength of the TS specimen was significantly reduced, compared to the NS and OS specimens. Compared to the strength of the NS specimen, the OS-1H specimen showed a 99% level, but the OS-2H specimen showed a 108% increase in strength. The OS-3H specimen decreased to a level of 48%, compared to the strength of NS. The TS-1H specimen was found to be 59% of the strength of NS. The TS-2H specimen was found to be 51% of the strength of NS, and the displacement increased compared to TS-1H. As with the OS-3H specimen, the TS-3H specimen decreased to 48% of the strength of NS. Therefore, in the case of a one-sided fire, the load ratio was in the range of 1.0, but in the case of a two-sided fire, it was found that the load-bearing capacity was lowered by more than 40–50%, compared to the non-heated state. The above load–displacement test results show that the fire design method according to the fire scenario should be considered, as shown in Table 3 of Eurocode [17]. In Eurocode’s Table 3, when the axial force ratio is 0.70 ($u_{fi}$ = 0.70) than the axial force ratio is 0.35 ($u_{fi}$ = 0.35), the required wall thickness and cover thickness are increased. In addition, the required wall thickness and cover thickness increased when the fire side exposed to the wall was on two-sided heating rather than one-sided heating. This means that the more surfaces exposed to the fire, the more the strength of the member is required. In the results of the residual strength test (Table 11), the test specimen with two-sided heating (TS) decreased more than twice as much like that of one-sided heating (OS). It shows that even if the structure cools sufficiently after exposed to fire on two sides or more, it does not recover as much as its original strength. From the results of this study’s full-scale experiment and Eurocode’s Table 3, it was confirmed that in the two-sided heating situation, the size and cover thickness of the member should be sufficiently secured rather than one-sided heating. In the 3 h fire scenario, the strength of the one-sided and two-sided heating specimens decreased by 52% compared to the non-heating specimen. Unlike the 3 and 4 h fire resistance standards (REI 180, 240) presented in Table 3 (Eurocode), there was no significant difference between the one-sided and two-sided fire scenarios in the 3 h experiment. Therefore, in a fire for a long time of 3 h or more, it is necessary to secure sufficient size and strength of members regardless of the fire scenario. In the wall (Table 6) and slab (Table 7) of ACI 216.1M-14 [18], the thickness of the member or the cover thickness design according to the fire scenario is not considered. In the future performance-based fire design, the criteria that consider the fire-exposed surface should be reflected in consideration of the fire scenario.

### 3.3. Moment–Deflection Angle Relationship

The moment and deflection angle of a structure are important properties to understand the strength and ductility capacity. Therefore, the moment–deflection angle relationship of the WSC one-sided or two-sided heating specimen and the difference in the deflection angles of the heated and non-heated surfaces were compared. Figure 15 shows a schematic of the WSC specimen to derive the relationship between the moment (M) and the deflection angle (*θ*), where the maximum moment ($M_{max}$) is the product (P×l) of the force (P, kN) and the distance (l, m) from the support to the actuator, and the deflection angle (*θ*) is given by Equation (6), according to the vertical LVDT 3 and 4. The deflection angle (*θ*) is the arc tangent relationship between the distance (l) from the support end to the LVDT 3 and 4, and the deformation (δ) of the LVDT 3 and 4 is derived by experiments. In each graph, the non-heating surface of the specimen is indicated as (N), and the heating surface is indicated as (H).
(6)$$tan{\left(\delta /l\right)}^{-1}=\theta$$

#### 3.3.1. Non–Heating Specimen (NS)

Figure 16 shows the graph of the moment–deflection angle relationship of the non-heating specimen (NS), where the location of each LVDT is as shown in Figure 8, LVDT-3 for Left and LVDT-4 for Right. Since NS is a non-heating specimen, Zones 1 and 2 in Figure 7 were not heated, and both Left and Right were marked as (N). In the graph of Figure 16, the location of the maximum moment at the maximum deflection angle is indicated as a symbol. The maximum moment ($M_{max}$) of NS is 1,276 kN·m, and the maximum left and right deflection angles ($\theta_{max}$) are (0.0284 and 0.0245) rad, respectively. Since the NS specimen was tested at room temperature (RT) without heating, as shown in Figure 16, there was no significant difference in the left and right deflection angles.

#### 3.3.2. One–Sided Heating Specimen (OS)

Figure 17 shows the graph of the moment–deflection angle relationship of the one-sided heating specimen (OS), where the vertical LVDT-3 is located about 100 mm from the wall of the unheated surface, and is marked as left (N). The vertical LVDT-4 is located about 100 mm from the wall of the heating surface, and is marked as right (H). The maximum moment ($M_{max}$) of OS-1H is 1261 kN·m, and the maximum left and right deflection angles ($\theta_{max}$) are (0.0159 and 0.0155) rad, respectively. The difference between the two deflection angles is 0.0004 rad. There was spalling effect on the heated surface of OS-1H, and some concrete of the wall fell off, but Figure 17a shows that there was no significant difference between the deflection angle of the unheated surface (left, N), and that of the heated surface (right, H). Cracks occurred in the connection of the heated surfaces, and the cracks progressed to the wall and the slab. The maximum moment ($M_{max}$) of OS-2H is 1382 kN·m, and the maximum left and right deflection angles ($\theta_{max}$) are (0.0201 and 0.0165) rad, respectively. The maximum moment ($M_{max}$) of OS-2H is 1382 kN·m, and the maximum left and right deflection angles ($\theta_{max}$) are (0.0201 and 0.0165) rad, respectively. Figure 17b shows that there was no significant difference between the deflection angles of the unheated connection (left, N), and the heated connection (right, H), until the initial 721 kN·m of OS-2H. However, when the specimen failed, the heated connection (right, H) was destroyed at 0.0165 rad with cracks, while the non-heated connection (left, N) was destroyed at 0.0201 rad. The difference between the two connection deflection angles was 0.0036 rad. This indicates that the ductility capacity of the non-heating connection part is greater than that of the heated connection part. Therefore, as the fire effect increases, the ductility decreases. The maximum moment ($M_{max}$) of OS-3H is 614 kN·m, and the maximum left and right deflection angles ($\theta_{max}$) are (0.0189 and 0.086) rad, respectively. Figure 17c shows that in OS-3H, there was a difference in the deflection angle between the unheated connection (left, N), and the heated connection (right, H) from the initial part. At 50% of the strength of the OS-3H specimen (moment strength 307.51 kN·m), the deflection angle of the heated connection (right, H) was 0.0021 rad, and the deflection angle of the unheated connection (left, N) was 0.0046 rad. The difference between the deflection angles of the two connections was about 0.0025 rad. At the maximum strength (moment 614.67 kN·m) of the OS-3H specimen, the deflection angle of the heated connection (right, H) was 0.0086 rad, and the deflection angle of the unheated connection (left, N) was 0.0189 rad. The difference in deflection angle between the two connections was 0.0103 rad, and the difference in deflection angle was more than four times larger than that of 50% strength (307.51 kN·m). In the case of the OS-1H and OS-2H specimens, the difference between the unheated connection (left, N) and the heated connection (right, H) was (0.0004 and 0.0036) rad, respectively, but in the case of the OS-3H specimen, it increased significantly to 0.0103 rad. In the case of the OS-3H specimen, the deflection angle of the heated connection (right, H) decreased with cracking, and the strength significantly decreased, compared to the non-heating connection (left, N).

#### 3.3.3. Two–Sided Heating Specimen (TS)

Figure 18 summarizes the graph of the moment–deflection angle relationship of the two-sided heating specimen (TS), where both vertical LVDT-3 and 4 are located about 100 mm from the wall of the heating surface. Since the two LVDTs are heating conditions, they are marked as left (H) and right (H), respectively. Figure 18a shows that the maximum moment ($M_{max}$) of TS-1H was 748 kN·m, and the maximum left and right deflection angles ($\theta_{max}$) were both 0.0043 rad. There was no difference in the deflection angle of the connections (left and right) of the two heating surfaces. Figure 18b shows that the maximum moment ($M_{max}$) of TS-2H is 647 kN·m, and the maximum left and right deflection angles ($\theta_{max}$) are (0.0107 and 0.0104) rad, respectively. The difference between the deflection angles of the two connections was 0.0003 rad. Figure 18c shows that the maximum moment ($M_{max}$) of TS-3H is 614 kN·m, and the maximum left and right deflection angles ($\theta_{max}$) are (0.0104 and 0.0089) rad, respectively. The difference between the two connection deflection angles was 0.00147 rad. At first, the difference in deflection angle of the left and right connections was not large, but the difference in deflection angle gradually occurred. In the case of the TS-1H and TS-2H specimens, there was no significant difference in the deflection angle of the left and right connections, as in the OS-1H and OS-2H specimens. However, in the case of the TS-3H specimen, there was a difference in the deflection angle of the left and right connections by more than 5 times compared to the TS-2H (0.00147 rad/0.00027 rad = 5.4). The moment was reduced by more than 18% compared to TS-1H.

#### 3.3.4. Result of the Moment–Deflection Angle

Table 12 summarizes the results of the moment–deflection angle of NS, OS, and TS. In addition, the difference in the moment ratio ($M/M_{NS}$) to the moment ($M_{NS}$) of the non-heating (NS) specimen and the deflection angles of the left and right connections was derived. There was a difference in moment and deflection angles according to the fire scenario. As a result of comparing the moment ratio (Table 12), the moment ratio of the TS specimen was reduced from 3 h to a maximum of 52%, compared to the NS and OS specimens. As a result of comparing the deflection angles, there was no difference in the left and right deflection angles of the OS-1H and TS-1H specimens. However, the difference between the left and right deflection angles of the OS-3H specimen was 0.01029 rad, and the difference of the TS-3H specimen was 0.00147 rad. Compared to the TS-3H specimen, the difference in the deflection angle of the OS-3H specimen occurred more than 7 times (0.01029 rad/0.00147 rad = 7). It means that the difference between the deflection angle between the non-heated side (Zone 2) and the heated side (Zone 1) occurred in the one-sided heating fire scenario. The two-sided heating test result showed a lower moment ratio and deflection angle than the one-sided heating test result, but the difference in left and right deflection angle was not large. Therefore, the performance of the wall-slab connection should be considered according to the fire scenario.

## 4. Comparison of the Experimental Results and Standards

Chapter 5 examines the simple calculation method (SCM) presented by the standards (Eurocode, NIST, and ACI). In addition, it compares the theoretical design fire moment derived for each standard and the WSC fire test results. The results can be used as basic data for proper repair and retrofit according to the fire scenario when fire scenarios occur on one or two sides through comparison of the theoretical design fire moment of the wall–slab and the fire test results.

### 4.1. Eurocode’s Simplified Calculations Method

Eurocode’s EN1992-1-2 Annex E Section E.2 (hereinafter, E.2) [17] presents a ‘simplified calculations method’ and an ‘alternate simplified calculation method’ for simple supports and slabs in fire situations. In the fire resistance design method according to the SCM, the section resistance moment ($M_{Rd,fi}$) at the position of maximum deflection moment can be calculated by the applied moment ($M_{Ed}$) E.2 [17] in Equations (7)–(9), where the value of $A_{s,prov}/A_{s,req}$ should not exceed 1.3, $A_{s,prov}$ means the cross-sectional area of the tensile rebar, and $A_{s,req}$ means the cross-sectional area of the tensile rebar required at room temperature (RT), as suggested in EN 1992-1-1. $\gamma_{s}$ is the partial material factor of EN 1992-1-1, and $\gamma_{s,fi}$ is the partial material factor in fire situations. $k_{s}\left(\theta \right)$ is the yield strength reduction factor of the rebar with temperature.
(7)$$M_{Ed,fi}\leq M_{Rd,fi}$$
(8)$$M_{Ed,fi}=\omega_{Ed,fi}\times l_{eff}{}^{2}/8$$
(9)$$M_{Rd,fi}=\left(\gamma_{s}/\gamma_{s,fi}\right)\times k_{s}\left(\theta \right)\times M_{Ed}\left(A_{s,prov}/A_{s,req}\right)$$

In addition, Eurocode proposes the ‘500 °C isotherm method (EN 1992-1-2 Annex B Section B.1)’ and ‘zone method (EN 1992-1-2 Annex B Section B.2)’ as an alternate simplified calculation method. Figure 19 shows that in the 500 °C isotherm method, it is assumed that the area where the concrete exceeds 500 °C is damaged, and does not contribute to the member performance. The member performance can be expressed as $M_{u}$, as shown in Equation (10), by using the sum of $M_{u1}$ and $M_{u2}$, as in the equivalent stress block in Figure 20. $M_{u1}$ can be obtained through the area and strength of the tensile rebar, as shown in Equation (11), and $f_{sd,fi}\left(\theta_{m}\right)$ is the yield strength for the average temperature ($\theta_{m}$) of the tensile rebar. Equation (12) shows that $M_{u2}$ is related to the area and strength of the compression rebar, and $f_{scd,fi}\left(\theta_{m}\right)$ is the yield strength for the average temperature ($\theta_{m}$) of the compression rebar, where $A_{s1}$ and $A_{s2}$ mean the cross-sectional area of the tensile rebar.
(10)$$M_{u}=M_{u1}+M_{u2}$$
(11)$$M_{u1}=A_{s1}f_{sd,fi}\left(\theta_{m}\right)z$$
(12)$$M_{u2}=A_{s2}f_{scd,fi}\left(\theta_{m}\right)z^{\prime }$$

The zone method is more complicated to calculate than the 500 °C isotherm method, but it can derive much more accurate results. The zone method is similar to the 500 °C isotherm method, in that it calculates by reducing the section by considering the damage of the concrete exposed to fire, but the difference is in consideration of the damage area of the concrete and the average temperature, average compressive strength, and elastic modulus. The zone method is similar to the 500 °C isotherm method, because it calculates by reducing the section by considering the damage of concrete exposed to fire, but it differs in consideration of the damage area of concrete, average temperature, average compressive strength, and elastic modulus. In this study, the fire design moment strength of WSC was derived using the 500 °C isotherm method, excluding the area over 500 °C in Table 10 and Figure 10.

### 4.2. Nist’s Simplified Calculations Method

In NIST (NIST Technical Note 1681) [19], the bending moment of a simply supported slab or beam can be calculated using the following Equations (13) and (14). In general, the concrete at the top of the beam or slab is not exposed to fire, and its strength does not decrease significantly, because the temperature is relatively low compared to the bottom. However, when the upper and lower surfaces of the slab or beam are exposed to fire, or when fire damage to the upper part of the member is expected, due to excessive fire, the reduction of concrete strength (${f^{\prime }}_{c}$) should be considered, where $A_{s}$ is the cross-sectional area of the tensile rebar, and $f_{s\theta }$ is the yield strength of the rebar at the fire-exposure temperature (θ).
(13)$$M_{n\theta }=A_{s}f_{s\theta }\left(d-a_{\theta }/2\right)$$
(14)$$a_{\theta }=A_{s}f_{s\theta }/0.85{f^{\prime }}_{c}b$$
where it is necessary to determine the effective fire-exposure depth ($u_{e}$) of the rebar, in order to derive the strength reduction factor of the rebar according to the temperature. If the rebar is at several distances from the fire-exposed surface, the effective depth of the rebar can be determined using the following Equation (15). In the case of rebar at the edge of the member, the effective distance can be calculated by the average of the distances of the two fire-exposure surfaces. Since the WSC specimen has a fixed rebar cover thickness of 40 mm, the calculation of Equation (15) is omitted.
(15)$$u_{e}=\sum_{i=1}^{n}u_{i}A_{i}/\sum_{i=1}^{n}A_{i}$$

### 4.3. ACI’s Simplified Calculations Method

ACI (ACI 216.1M-14) [18], like Eurocode, suggests a simple procedure method for evaluating the fire resistance performance of simple supports and slabs. In ACI, the fire resistance performance of a member can be determined according to the aggregate. For example, for a siliceous aggregate, such as this WSC specimen, the fire resistance performance of the member can be determined using a graph, as shown in Figure 21, where ω can be calculated by Equation (16). Unlike Eurocode [17], ACI [18] does not separately consider the strength reduction factor of rebars, and can obtain the fire moment through the ratio of the bending moment ($M_{n}$) to the fire moment (M) at RT, where u represents the average thickness (mm) between the center of the main rebar and the concrete fire-exposed surface.
(16)$$\omega =A_{s}f_{y}/bd{f^{\prime }}_{c}$$

The simple design method proposed by ACI [18] is a method of deriving the moment in the case of fire by using the moment ratio as a specification method. The method of obtaining the positive moment is presented as Equation (17), which is the same as Equation (13) of NIST [19], where $A_{s}$ is the cross-sectional area of the tensile rebar, and $f_{y\theta }$ is the strength reduction of the rebar according to temperature. Since the method of calculation by substituting the residual strength of a material (concrete, rebar) according to temperature is the same as Equation (13) presented by NIST, the calculation results of ACI [18] and NIST [19] are presented together.
(17)$$M_{n\theta }{}^{+}=A_{s}f_{y\theta }\left(d-a_{\theta }/2\right)$$

### 4.4. Comparison of Simplified Calculation Methods and Experimental Results

Table 13 compares and organizes the design loads and moments derived by the SCM for each standard (Eurocode, ACI, and NIST) and the WSC test results. The WSC boundary condition was assumed to be simply supported. Figure 15 presents the conditions and dimensions of the support, while Table 4 presents the specimen information, where $P_{test,OS}$ and $M_{test,OS}$ are the loads and moments of the one-sided heating (OS) test, and $P_{test,TS}$ and $M_{test,TS}$ are the loads and moments of the two-sided heating (TS) test, respectively. For each load ($P_{EC}$, $P_{ACI,NIST}$) and moment ($M_{EC}$, $M_{ACI,NIST}$), Equations (10–12) presented by Eurocode and Equations (13) and (14) presented by ACI and NIST were used. In the equation for each standard, the strength reduction factor ($k_{f,Test}$, $k_{f,EC}$, $k_{f,ACI}$) in Table 10 was substituted and derived. The experimental strength reduction factor ($k_{f,Test}$, column D) and Eurocode strength reduction factor ($k_{f,EC}$, column E) of Table 10 were entered into Equations (10–12) of Eurocode, and summarized as load ($P_{EC}$) and moment ($M_{EC}$), respectively. In addition, the experimental strength reduction factor ($k_{f,Test}$, column D) and ACI strength reduction factor ($k_{f,ACI}$, column F) of Table 10 were entered into Equations (13) and (14) of ACI and NIST, and summarized as load ($P_{ACI,NIST}$) and moment ($M_{ACI,NIST}$), respectively.

Figure 22 compares and summarizes in a graph the experiment and the load (P, kN) by standard. As a result of comparison between the OS ($P_{Test,OS}$) and TS test ($P_{Test,TS}$), a difference in strength of up to 1025 kN occurred in 2 h of heating. In Zone A (blue line) of Figure 22, the OS test ($P_{test,OS}$) was closer to the Eurocode graph ($P_{EC}$) than the ACI and NIST graphs ($P_{ACI,NIST}$) for (0 to 2) h of heating. However, Figure 22 shows that the OS test ($P_{test,OS}$) 3 h was located in the ACI and NIST graphs ($P_{ACI,NIST}$) and Zone B (purple line) area. All of the TS tests ($P_{test,TS}$) for (1 to 3) h of heating were located in Zone B, and were similar to the graphs of ACI and NIST ($P_{ACI,NIST}$).

Figure 23 compares and summarizes in a graph the experiment and the moment (M, kN·m) by standard. As a result of comparing the moment ($M_{test,OS}$) of OS and the moment ($M_{test,TS}$) of the TS heating test, a difference of up to 735 kN·m occurred in 2 h of heating. In Zone A (blue line) of Figure 23, the OS test was closer to the Eurocode graph ($M_{EC}$) than the ACI and NIST graphs ($M_{ACI,NIST}$) for (0 to 2) h of heating. However, Figure 23 shows that the OS test ($M_{test,OS}$) 3 h was located in the ACI and NIST graphs ($M_{ACI,NIST}$) and Zone B area (purple line).

The following conclusions were drawn through the results of the load comparison graph in Figure 22, and the moment comparison graph in Figure 23. (1) The calculation results of Eurocode showed similar results to the one-sided heating (OS) test. (2) The calculation results of ACI and NIST were similar to those of the two-sided heating (TS) test. (3) In the 2 h heating test, the differences between the load and the moment of the OS test and the TS test results were up to 1025 kN and 735 kN·m, respectively, and the greatest difference occurred compared to other heating times. (4) As a result of comparing Zones A and B, in a fire scenario in which a one-sided heating situation occurred, the strength of the connection after the fire can be inferred according to the Eurocode method. (5) In a fire scenario where a two-sided heating situation occurs, the strength of the connection after the fire can be inferred according to the method of ACI and NIST. (6) The strength decreased rapidly, due to the spalling and cracking of the concrete cooled at high temperature (600 °C) (Table 1). In this way, the WSC specimens (OS-3H, TS-3H) heated for more than 600 °C for 3 h had spalling, and many microcracks (Figure 9), and the residual performance rapidly decreased. However, since the rebar strength recovers at high temperatures (above 600 °C), as shown in Table 2, it seems that the residual performance of the 3 h WSC specimens was maintained by more than 50%.

## 5. Conclusions

In this study, a full-scale specimen of the RC wall–RC slab connection (WSC) was planned, and experimental works were performed to evaluate both the residual strength and moment in accordance with the fire scenarios, including heatings with one side and two sides. The residual strength of concrete and rebar were derived through the theories and cooling tests. In addition, the calculation results of the maximum load and moment by the standards of Eurocode, ACI, and NIST were adopted and compared with the residual strength in terms of 1, 2, and 3 h fire scenarios. Through this study, the following results were derived:As compared with the non-heating specimen, in 1 h heating, the one-sided heating specimen showed no decrease in strength. In 2 h heating, the strength increased by more than about 8%. However, at 3 h heating, the strength decreased notably by over 52%. In the case of 2 h heating, the temperature of the slab rebar was about 245 °C, and the temperature of the connection rebar was 445 °C. It was identified that the strength of rebars partially cooled at the temperatures of 200 to 500 °C recovered. Therefore, the strength of the one-sided heating specimen in 2 h of heating increased slightly compared to the room temperature specimen. However, the strength of the two-sided heating specimen was reduced by 41% in 1 h of heating, when compared to the NS. At 2 h of heating, strength decreased by more than 49%; and at 3 h of heating, the strength decreased by more than about 52% in sharp rate.Compared to the one-sided heating test, the strength of the two-sided heating test specimen decreased by more than 2 times in both 1 h and 2 h. However, in the 3 h fire, both the one-sided and two-sided test specimens showed a strength reduction by more than twice (52%), compared to the non-heating specimens. In the 3 h fire rates, the maximum load of one-sided, two-sided specimen and the maximum moment of one-sided, two-sided specimen were 856 kN and 614 kN·m, respectively. Therefore, in a 3 h fire, the strength decreased by more than 2 times compared to the non-heating specimens, regardless of the fire scenario (one-sided, two-sided). Therefore, in a fire for a long time of 3 h or more, it is necessary to secure sufficient size and strength of members regardless of the fire scenario.Compared to the two-sided fire scenario, there was a different deflection angle between fired Zone 1 and non-fired Zone 2 in the one-sided fire scenario. The difference between the left and right maximum deflection angles in one-sided specimen was about 0.01029 rad in a 3 h fire. On the other hand, the difference between the maximum left and right deflection angles in two-sided specimen was found to be 0.00147 rad in a 3 h fire. The maximum deflection angle of two-sided specimen was seven times greater than that of one-sided specimen. Compared to the two-sided specimen, it can be interpreted as the significant difference in the deflection angle could be occurred at the left and right connection areas in the one-sided specimen, due to the unbalance fire strength.As a result of comparing the fire tests with the simple calculation results of the standards (Eurocode, ACI) and NIST, the one-sided fire scenario is located in Zone A (see Figure 22 and Figure 23) well agreed to the results of Eurocode 1992-1-2. The two-sided fire scenario is located in Zone B (see Figure 22 and Figure 23), similar to the results of ACI 216 1M-14 (NIST 1681). The maximum load difference between the one-sided and two-sided tests was about 1025 kN in the 2 h heating, and the maximum moment difference was about 735 kN·m in the 2 h heating test.The maximum moments and strengths by Eurocode’s simplified calculation showed similar results to the one-sided heating test result. The simplified calculation results both ACI and NIST showed similar results to the two-sided heating test. As a result of comparisons between Zone A and Zone B in one side scenario, the residual strength at the connection after the fire can be inferred according to the Eurocode equation. In a fire scenario where a two-sided heating situation occurs, the residual strength of the connection after the fire can be inferred according to the equations of ACI and NIST. In the future performance-based fire design, the criteria that consider the fire-exposed surface should be reflected in consideration of the fire scenario.

## Figures and Tables

**Figure 1 materials-14-01793-f001:**
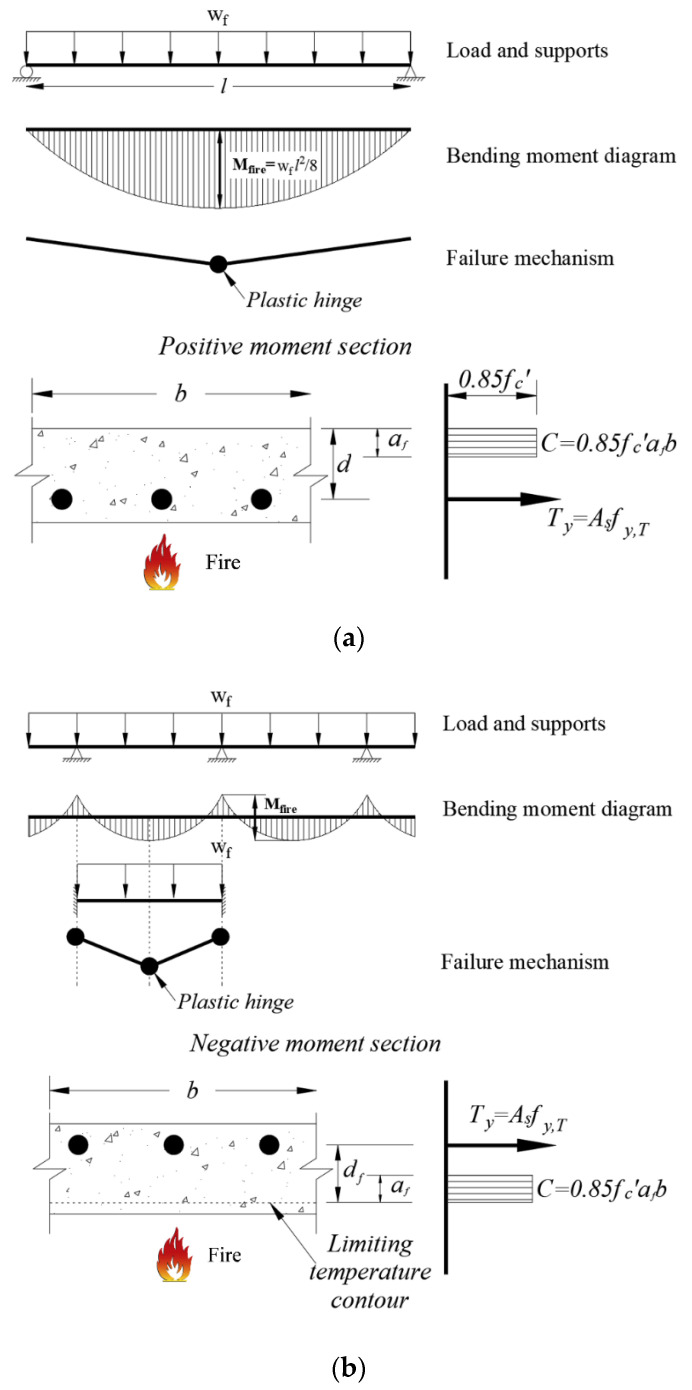
Failure mechanism and equivalent compressive stress block of reinforced concrete (RC) slab exposed to fire: (**a**) Simply supported; (**b**) continuously supported.

**Figure 2 materials-14-01793-f002:**
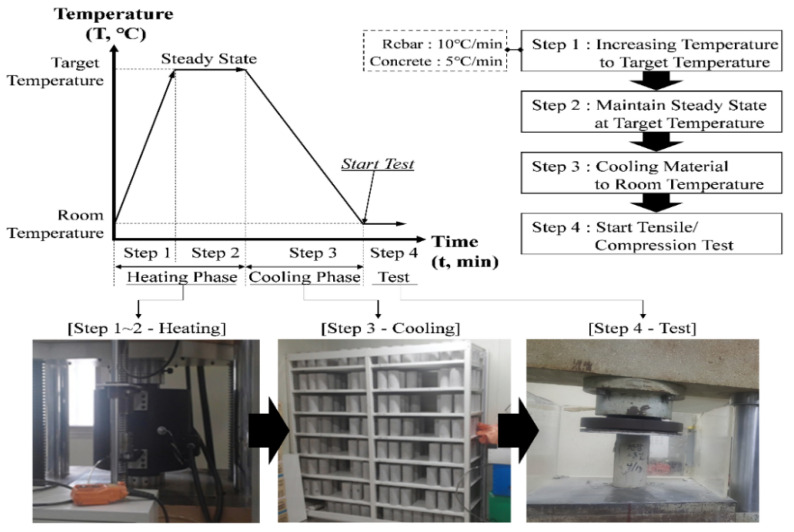
The procedure of cooling effect test (CIA, cooling in air).

**Figure 3 materials-14-01793-f003:**
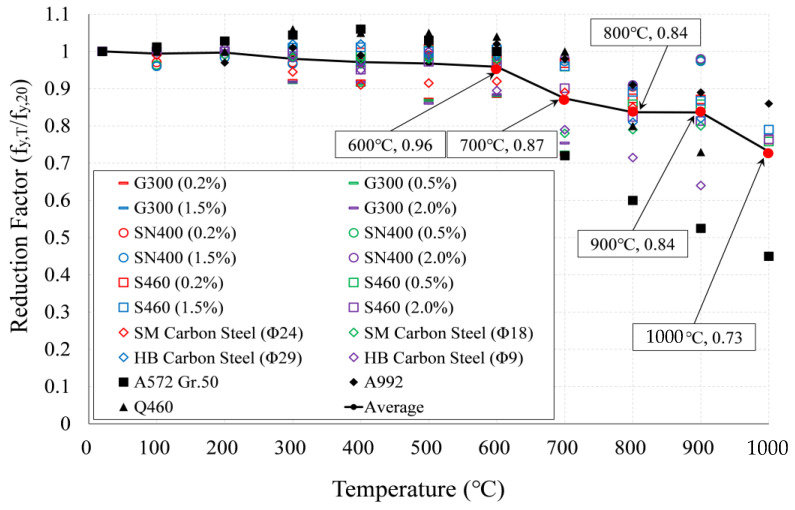
Comparison of reduction factor CIA test.

**Figure 4 materials-14-01793-f004:**
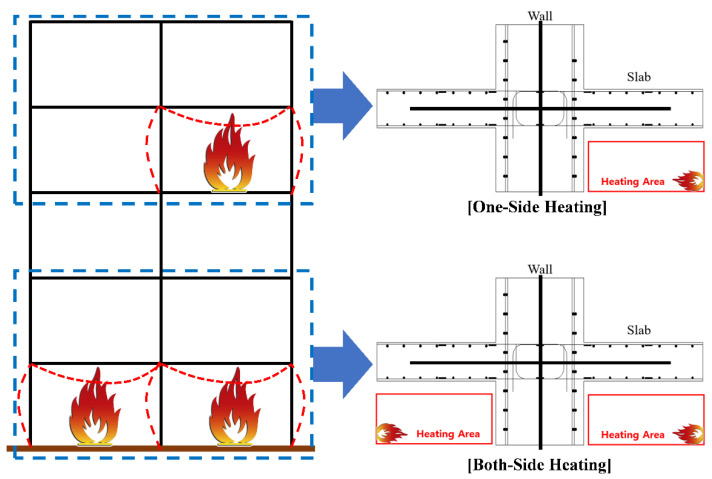
Concept of one-sided heating and two-sided heating.

**Figure 5 materials-14-01793-f005:**
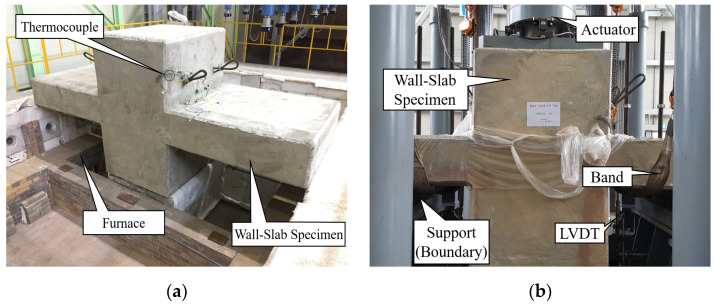
Wall-slab specimen test setting: (**a**) Heating furnace; (**b**) actuator.

**Figure 6 materials-14-01793-f006:**
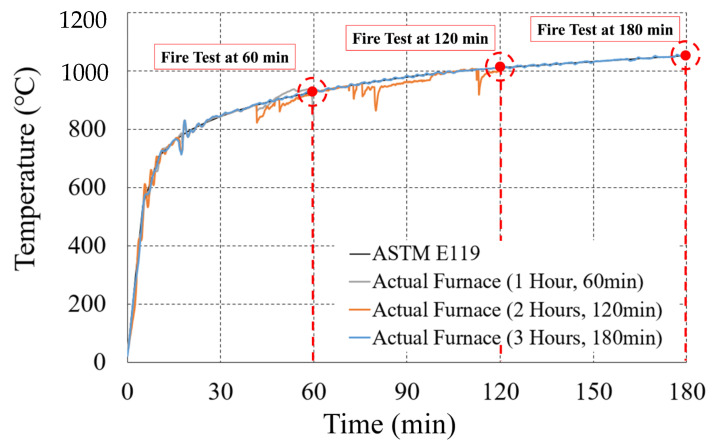
Time-temperature curve (ASTM E119-15 [35]).

**Figure 7 materials-14-01793-f007:**
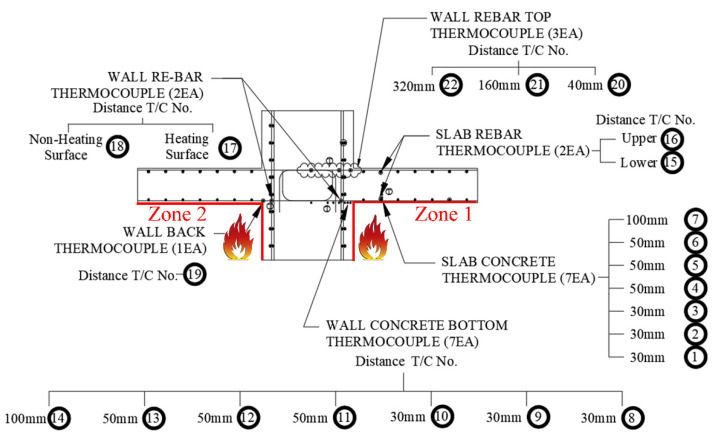
Location of the thermocouple.

**Figure 8 materials-14-01793-f008:**
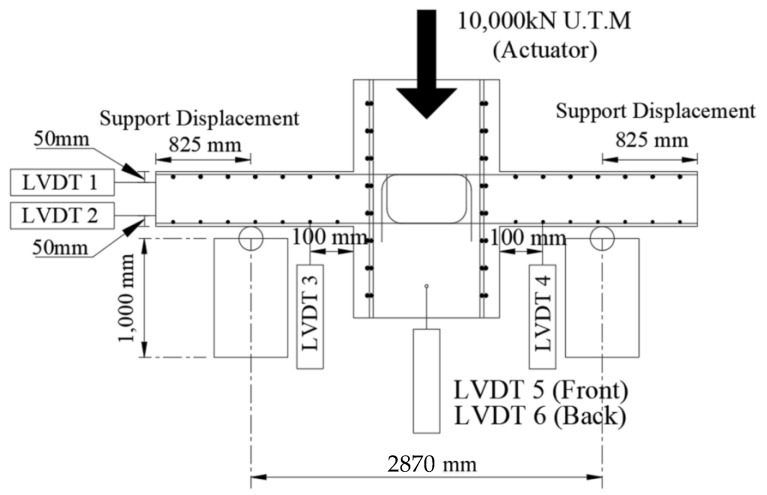
Location of LVDT.

**Figure 9 materials-14-01793-f009:**
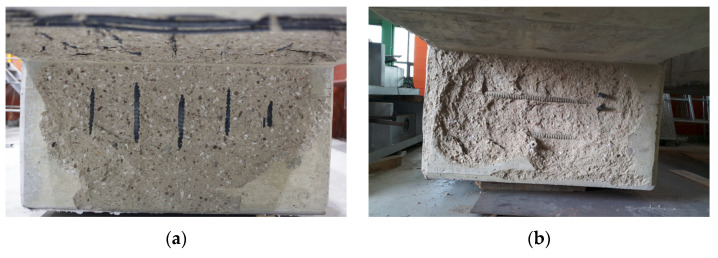
The exposed surface of wall-slab specimen after the fire test (3 h): (**a**) One-side heating (OS); (**b**) two-side heating (TS).

**Figure 10 materials-14-01793-f010:**
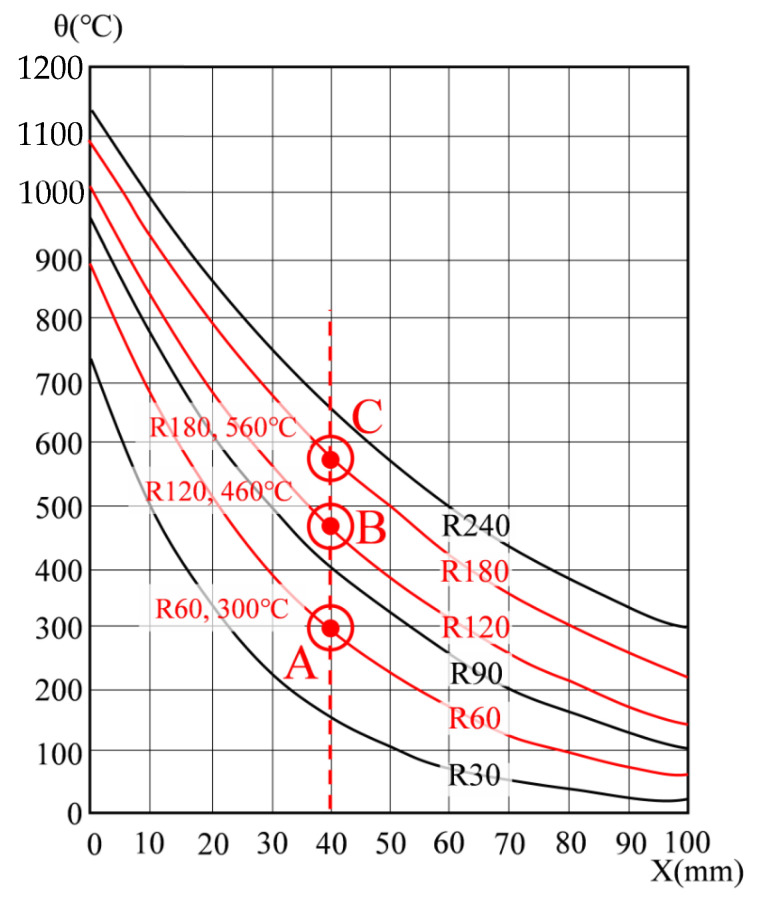
Temperature profiles for slabs (X = 40 mm) (EN 1992-1-2, reproduced).

**Figure 11 materials-14-01793-f011:**
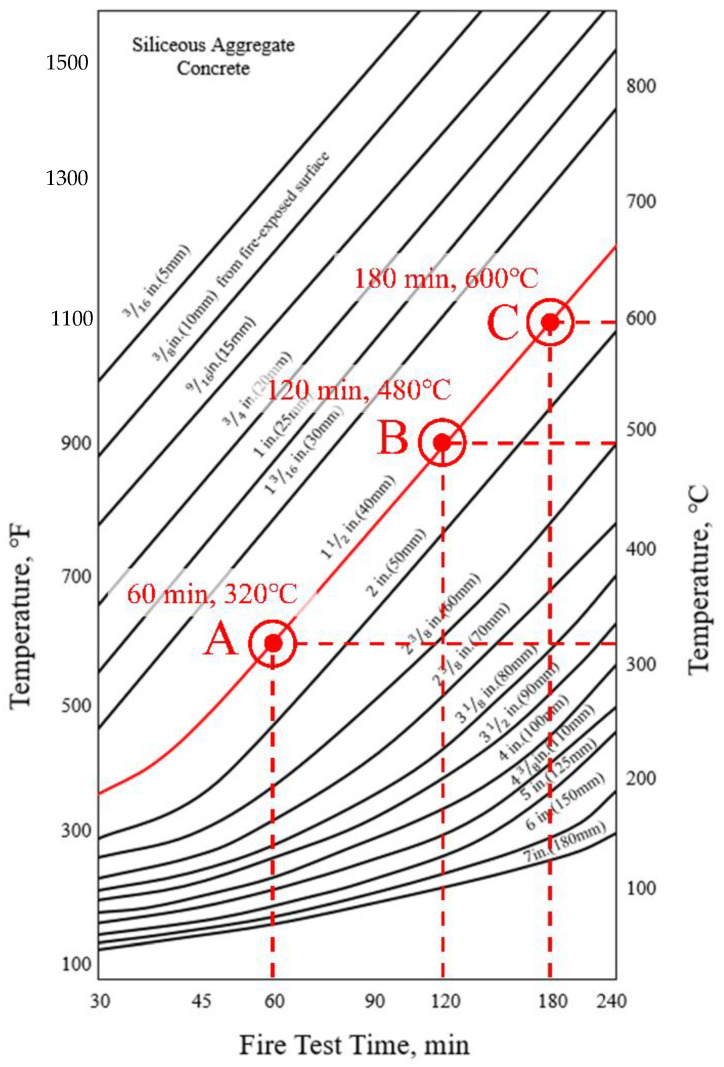
Temperature within slabs for siliceous aggregate concrete with ASTM E119.

**Figure 12 materials-14-01793-f012:**
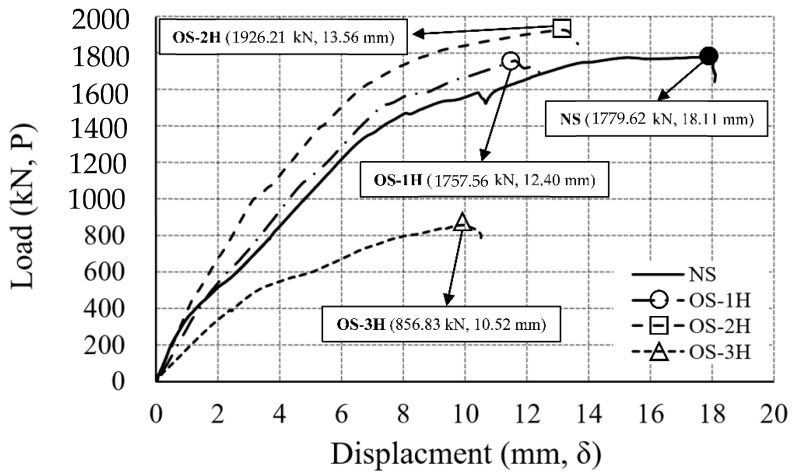
Load–displacement relationship (NS, OS specimen).

**Figure 13 materials-14-01793-f013:**
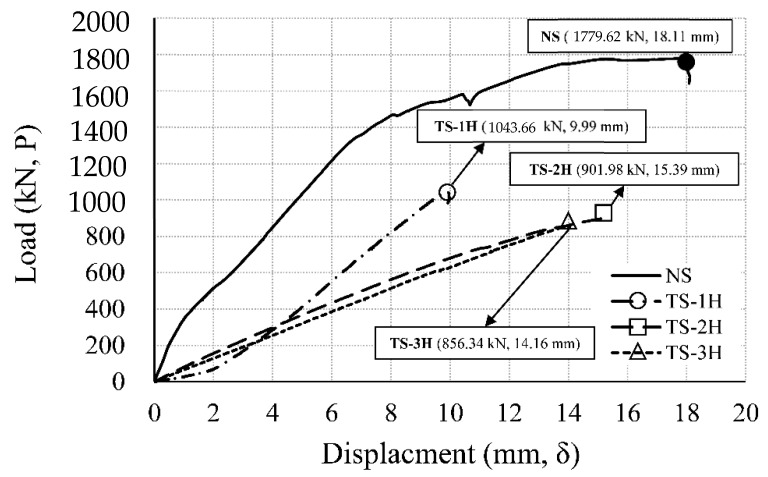
Load–displacement relationship (NS, TS specimen).

**Figure 14 materials-14-01793-f014:**
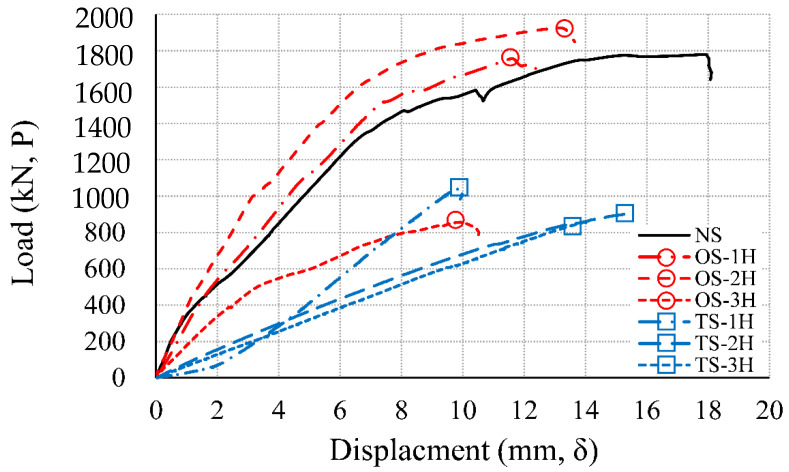
Comparison of load–displacement relationships (NS, OS, and TS specimen).

**Figure 15 materials-14-01793-f015:**
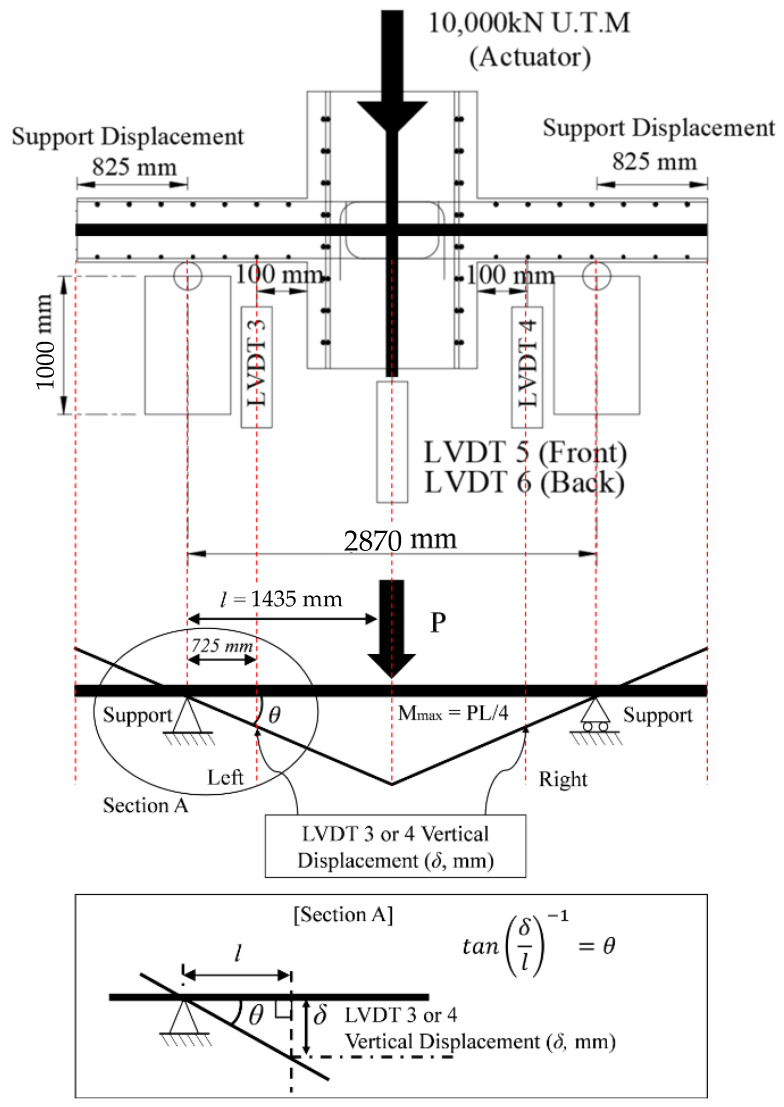
Schematic of the moment–deflection angle relationship.

**Figure 16 materials-14-01793-f016:**
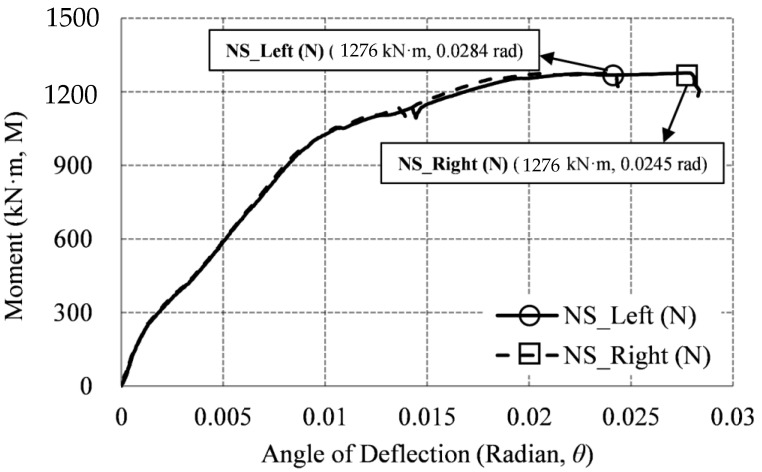
Moment–deflection angle relationship (NS specimen).

**Figure 17 materials-14-01793-f017:**
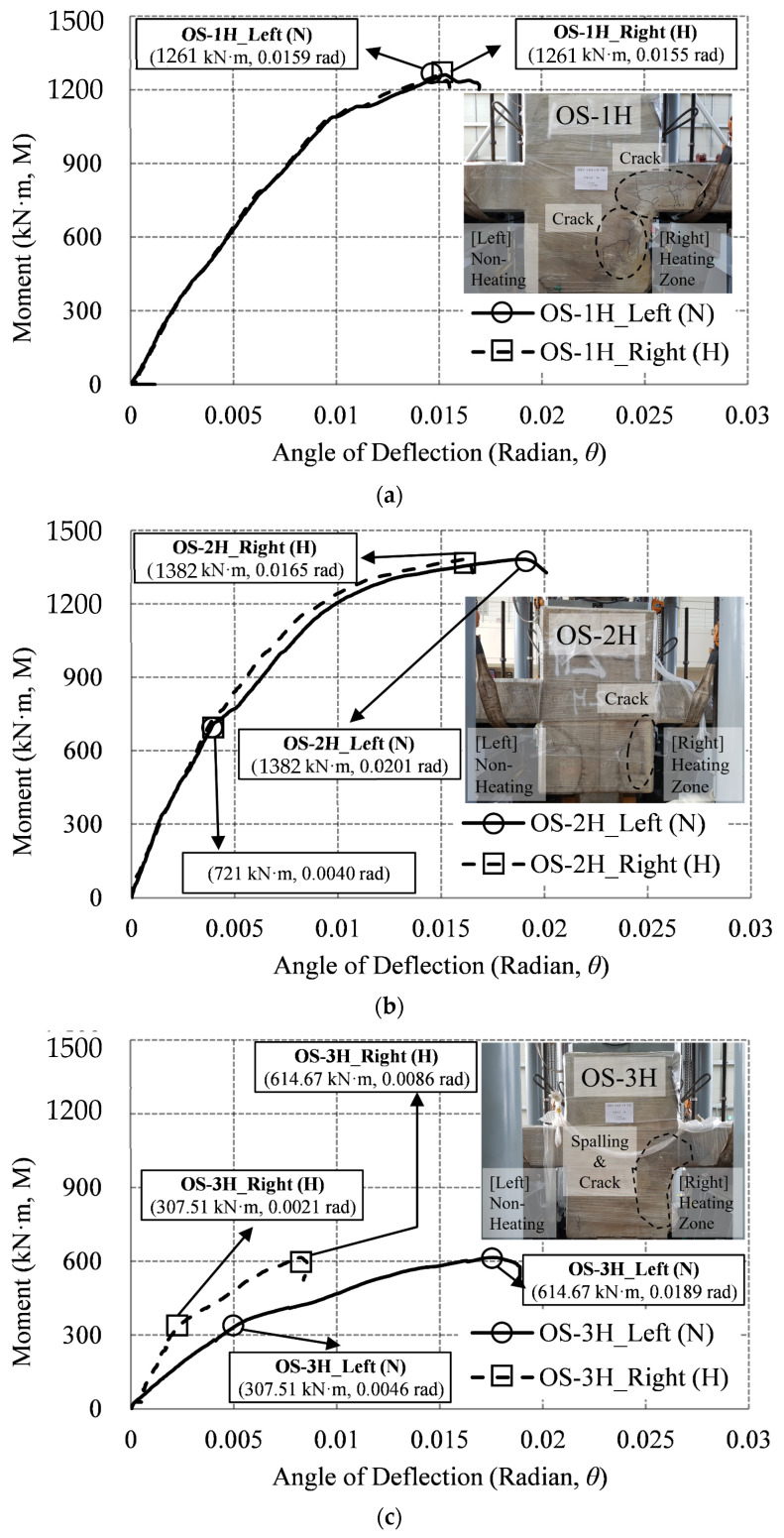
Moment–deflection angle relationship (OS specimen): (**a**) 1 h (OS-1H); (**b**) 2 h (OS-2H); (**c**) 3 h (OS-3H).

**Figure 18 materials-14-01793-f018:**
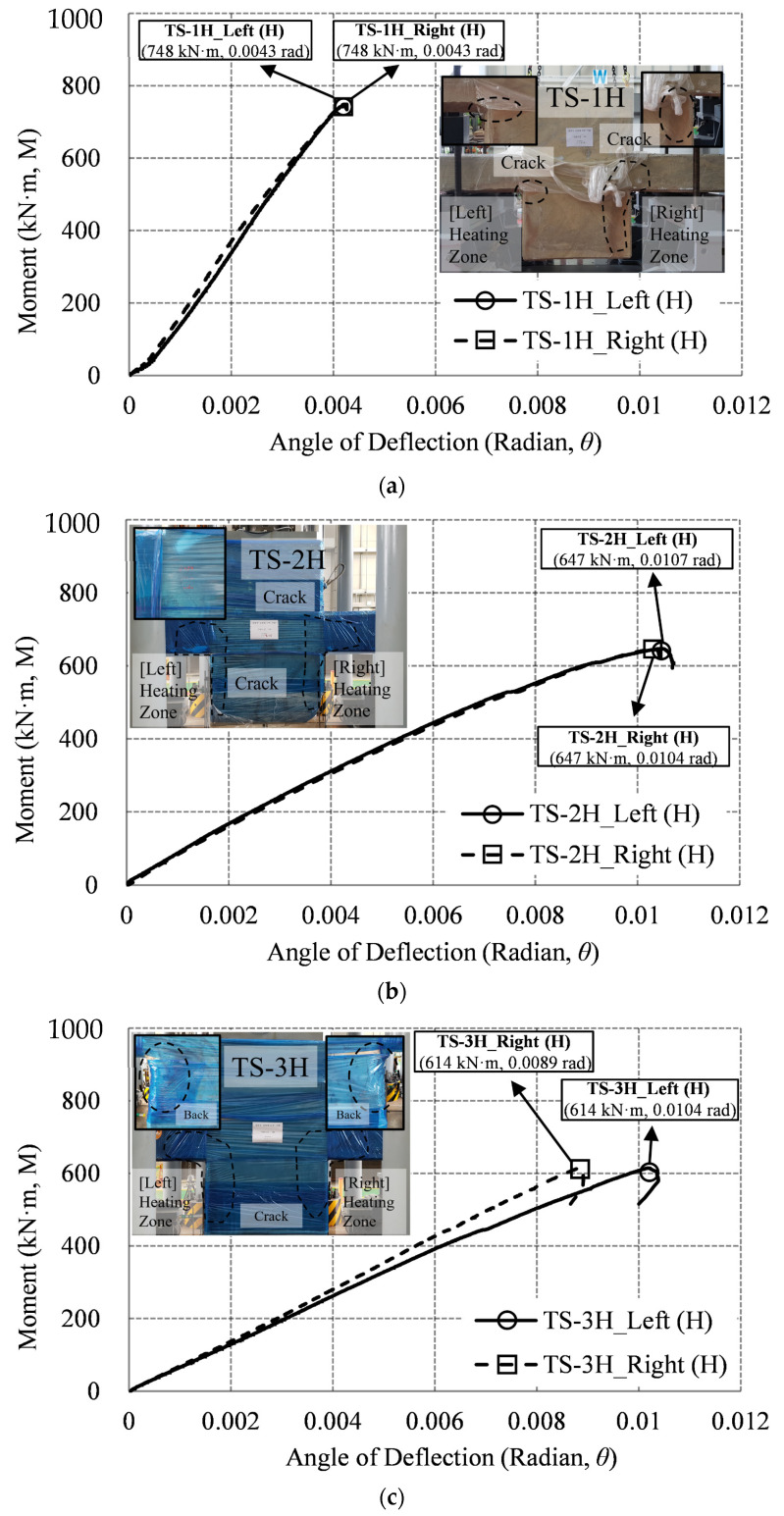
Moment–deflection angle relationship (TS specimen): (**a**) 1 h (TS-1H); (**b**) 2 h (TS-2H); (**c**) 3 h (TS-3H).

**Figure 19 materials-14-01793-f019:**
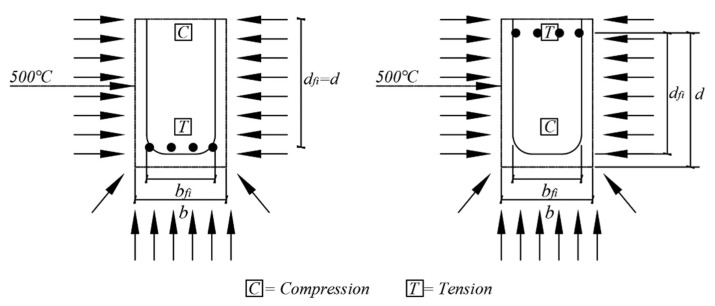
Reduced cross-section of reinforced concrete beam and column (EN 1992-1-2 Annex B, reproduced).

**Figure 20 materials-14-01793-f020:**
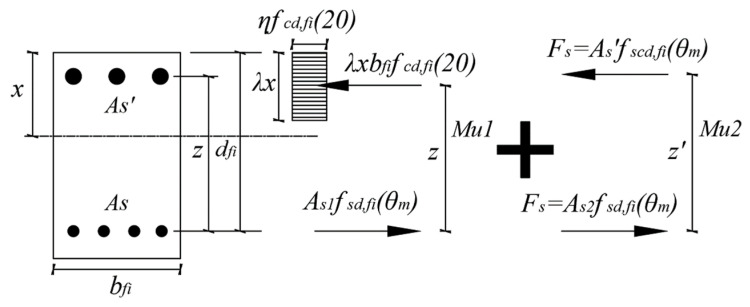
Stress distribution at ultimate limit state for a rectangular concrete cross-section with compression reinforcement (EN 1992-1-2 Annex B, reproduced).

**Figure 21 materials-14-01793-f021:**
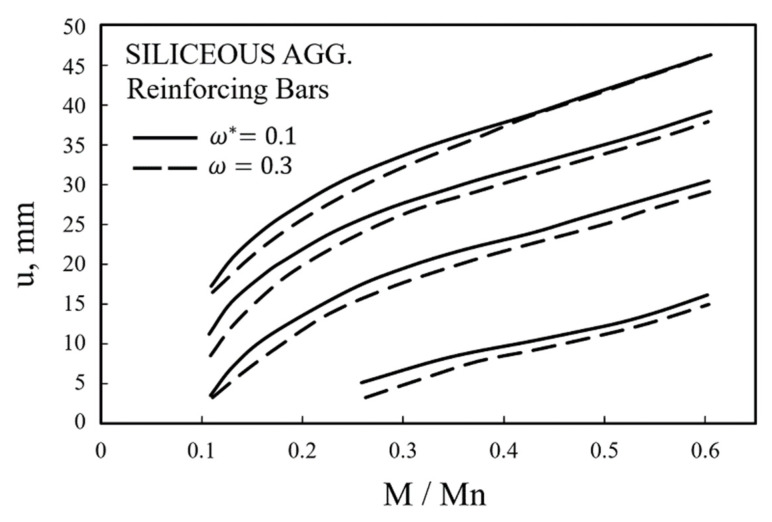
Simply supported beam and slab design (siliceous aggregates, reproduced).

**Figure 22 materials-14-01793-f022:**
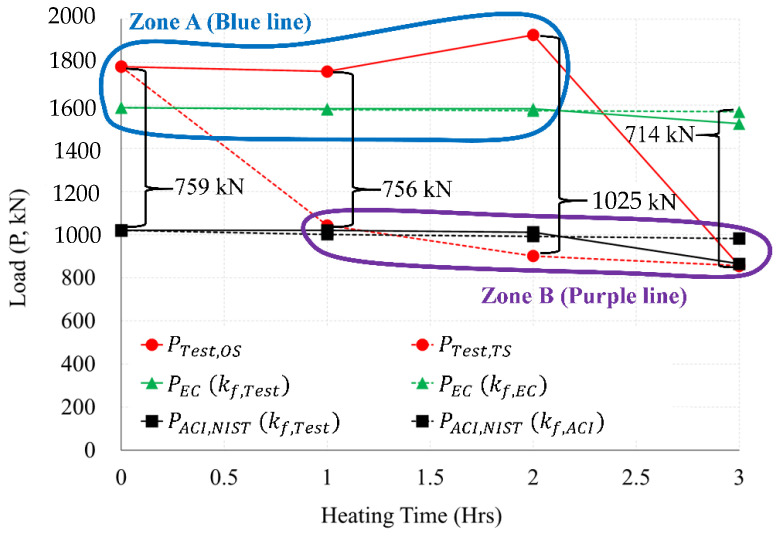
Comparison of load results by test (OS, TS) and Standard (EN1992-1-2, ACI 216.1M-14 and NIST 1681).

**Figure 23 materials-14-01793-f023:**
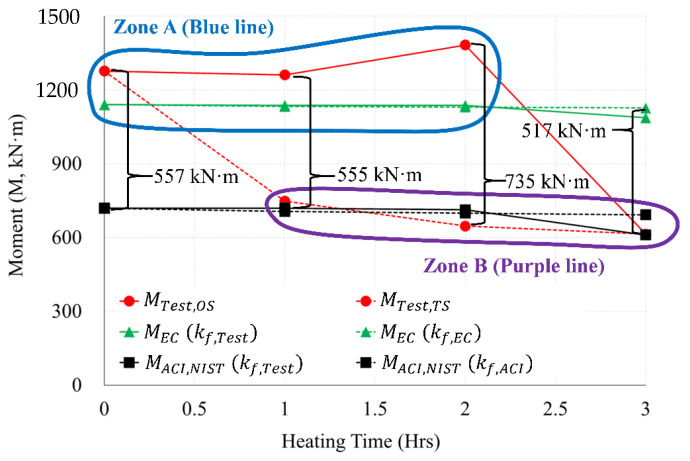
Comparison of moment results by test (OS, TS) and Standard (EN1992-1-2, ACI 216.1M-14 and NIST 1681).

**Table 1 materials-14-01793-t001:** Residual materials properties of siliceous concrete (35 MPa) after the fire.

Concrete Internal Temperature (°C, θ)	Compressive Strength				Elastic Modulus			
	$\mathrm{Compressive}\;\mathrm{Strength}\;(f_{c,\theta },\;\mathrm{MPa})$	Standard Deviation	$\mathrm{Average}\;\mathrm{Compressive}\;\mathrm{Strength}\;({f^{\prime }}_{c,\theta },\;\mathrm{MPa})$	$\mathrm{Reduction}\;\mathrm{Factor}\;\;({f^{\prime }}_{c,\theta }/{f^{\prime }}_{c,20})$	$\mathrm{Elastic}\;\;\mathrm{Modulus}\;(E_{c,\theta },\;\mathrm{MPa})$	Standard Deviation	$\mathrm{Average}\;\mathrm{Elastic}\;\;\mathrm{Modulus}\;({E^{\prime }}_{c,\theta },\;\mathrm{MPa})$	$\mathrm{Reduction}\;\mathrm{Factor}\;({E^{\prime }}_{c,\theta }$ $/{E^{\prime }}_{c,20})$
20	40.80	0.81	41.93 (${f^{\prime }}_{c,20}$)	1.00	24,200	927.36	23,700 (${E^{\prime }}_{c,20}$)	1.00
	42.40				22,400			
	42.60				24,500			
100	38.30	0.22	38.00	0.91	22,000	974.11	20,633	0.87
	37.90				20,100			
	37.80				19,800			
200	35.30	0.65	35.30	0.84	18,000	531.25	18,033	0.76
	36.10				17,400			
	34.50				18,700			
300	28.40	1.64	28.13	0.67	12,800	294.39	12,400	0.52
	26.00				12,300			
	30.00				12,100			
400	27.40	1.51	25.83	0.62	9900	402.77	9333	0.39
	26.30				9000			
	23.80				9100			
500	19.80	1.06	21.10	0.50	5300	355.90	5800	0.24
	22.40				6100			
	21.10				6000			
600	16.30	0.82	15.93	0.38	1700	163.30	1900	0.08
	16.70				2100			
	14.80				1900			
700	11.90	0.22	11.80	0.28	1100	124.72	1233	0.05
	11.50				1400			
	12.00				1200			
800	7.90	0.33	8.30	0.20	1000	47.14	933	0.04
	8.70				900			
	8.30				900			
900	4.20	0.28	3.80	0.09	500	47.14	433	0.02
	3.60				400			
	3.60				400			
1000	2.50	0.26	2.63	0.06	300	47.14	233	0.01
	3.00				200			
	2.40				200			

**Table 2 materials-14-01793-t002:** Reduction factor of mild steel after the fire.

Steel Temperature (°C, θ)	$\mathrm{Residual}\;\mathrm{Yield}\;\mathrm{Strength}\;\mathrm{Reduction}\;\mathrm{Factor}\;(f_{y,T}/f_{y,20})$																				Residual Elastic Modulus $(E_{y,\theta }/E_{y,20})$
	G300 [28] (Strain Level, %)				SN400 [29] (Strain Level, %)				S460 [30] (Strain Level, %)				Carbon Steel [31]				A572 Gr.50 [32]	A992 [21]	Q460 [30]	Average	Average
													**SM**		**HB**						
	0.2	0.5	1.5	2.0	0.2	0.5	1.5	2.0	0.2	0.5	1.5	2.0	Φ24	Φ18	Φ29	Φ9					
20	1.00	1.00	1.00	1.00	1.00	1.00	1.00	1.00	1.00	1.00	1.00	1.00	1.00	1.00	1.00	1.00	1.00	1.00	1.00	1.00	1.00
100	1.00	1.00	1.00	1.00	0.97	0.96	0.96	0.99	1.00	1.00	1.00	1.00	0.99	1.00	1.00	1.00	1.01	1.00	1.00	0.99	
200	1.00	1.00	1.00	1.00	0.98	0.98	0.98	0.99	1.00	1.00	1.00	1.00	0.99	0.99	1.01	1.01	1.03	0.97	1.00	1.00	
300	0.92	0.92	0.92	0.92	0.97	0.97	0.97	0.98	1.00	1.00	1.01	1.00	0.95	0.99	1.02	0.99	1.04	1.01	1.06	0.98	
400	0.92	0.91	0.92	0.92	0.97	0.97	0.97	0.98	1.00	0.99	1.01	0.95	0.91	0.98	1.02	0.95	1.06	0.99	1.05	0.97	
500	0.87	0.87	0.86	0.86	0.99	0.99	0.98	1.00	1.01	1.01	1.00	0.97	0.92	0.98	0.99	1.00	1.03	1.02	1.05	0.97	
600	0.88	0.88	0.88	0.88	0.99	0.98	0.97	0.99	0.98	0.97	1.01	0.97	0.92	0.98	0.99	0.90	1.00	1.02	1.04	0.96	
700	0.72	0.73	0.75	0.75	0.97	0.97	0.97	0.98	0.97	0.96	0.96	0.90	0.89	0.78	0.79	0.79	0.72	0.98	1.00	0.87	
800	-	-	-	-	0.90	0.91	0.91	0.91	0.87	0.86	0.89	0.82	0.85	0.79	0.81	0.72	0.60	0.91	0.80	0.84	
900	-	-	-	-	0.98	0.98	0.97	0.98	0.87	0.86	0.87	0.81	0.82	0.80	0.82	0.64	0.53	0.89	0.73	0.84	
1000	-	-	-	-	-	-	-	-	0.76	0.76	0.79	0.76	-	-	-	-	0.45	0.86	-	0.73	

**Table 3 materials-14-01793-t003:** Minimum dimensions and axis distances for load-bearing reinforced concrete walls (EN1992-1-2 [17]).

Standard Fire Resistance	Minimum Dimensions (mm) Wall Thickness/Axis Distance for			
	$u_{fi}=0.35$		$u_{fi}=0.70$	
	Wall Exposed on One Side	Wall Exposed on Two Sides	Wall Exposed on One Side	Wall Exposed on Two Sides
1	2	3	4	5
REI 30	100/10 *	120/10 *	120/10 *	120/10 *
REI 60	110/10 *	120/10 *	120/10 *	140/10 *
REI 90	120/10 *	140/10 *	140/25	170/25
REI 120	150/25	160/25	160/35	220/35
REI 180	180/40	200/45	210/50	270/55
REI 240	230/55	250/55	270/60	350/60

* Normally, the cover required by EN 1992-1-1 will control.

**Table 4 materials-14-01793-t004:** Section and mechanical properties of the wall-slab connection.

Section of Wall-Slab Connection			
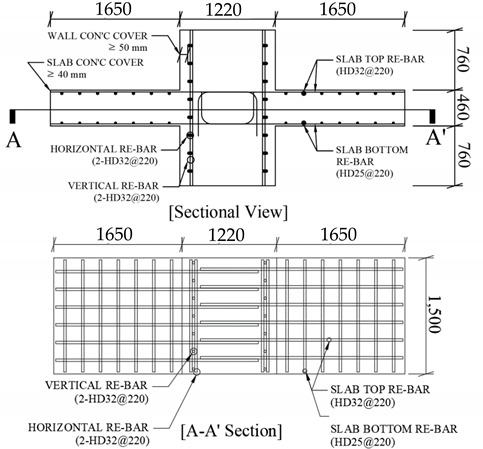			
**Material**	**Properties**		
**Rebar** **(SD400)**	**Yield,** $\;f_{y,20}$ **(MPa)**		400~520
	**Ultimate,** $\;f_{u,20}$ **(MPa)**		$f_{u,20}\geq 1.15f_{y,20}$
	**Diameter (mm)**	**Slab top, wall**	32 (D32)
		**Slab bottom**	25 (D22)
**Concrete**	**Compression,** $\;{f^{\prime }}_{c,20}$ **(MPa)**		35
	**Slump (mm)**		120
	**Aggregates type**		Siliceous

**Table 5 materials-14-01793-t005:** Minimum dimensions and axis distances for reinforced and prestressed concrete simply supported one-way and two-way solid slabs (EN1992-1-2 [17]).

Standard Fire Resistance	Minimum Dimensions (mm) Wall Thickness/Axis Distance for			
	$\mathrm{Slab}\;\mathrm{Thickness}\;h_{s}\left(mm\right)$	Axis-Distance		
		One Way	Two Way	
			$l_{y}/l_{x}\leq 1.5\;**$	$1.5\leq l_{y}/l_{x}\leq 2.0$
1	2	3	4	5
REI 30	60	10^*^	10 *	10 *
REI 60	80	20	10 *	15 *
REI 90	100	30	15 *	20
REI 120	120	40	20	25
REI 180	150	55	30	40
REI 240	175	65	40	50

* Normally, the cover required by EN 1992-1-1 will control. ** $l_{x}$ and $l_{y}$ are the spans of a two-way slab (two directions at right angles) where $l_{y}$ is the longer span.

**Table 6 materials-14-01793-t006:** Fire resistance of single-layer concrete walls, floors, and roofs (ACI 216.1M-14 [18]).

Aggregate Type	Minimum Equivalent Thickness for Fire-Resistance Rating, mm				
	1 h	1–1/2 h	2 h	3 h	4 h
Siliceous	90	110	125	155	175
Carbonate	80	100	115	145	170

**Table 7 materials-14-01793-t007:** Minimum cover in concrete floors and roof slabs (ACI 216.1M-14 [18]).

Aggregate Type	Minimum Equivalent Thickness for Fire-Resistance Rating, mm					
	Restrained	Unrestrained				
	4 or Less	1 h	1–1/2 h	2 h	3 h	4 h
**Non-Prestressed**						
Siliceous	20	20	20	25	30	40
Carbonate	20	20	20	20	30	30
**Prestressed**						
Siliceous	20	30	40	45	60	70
Carbonate	20	25	35	40	55	55

**Table 8 materials-14-01793-t008:** Wall-slab connection specimen test plan.

Specimen	Fire Scenario	Fire Curve	Heating Time	Manufacture (EA)
NS	Non-heating	ASTM E119	0	1
OS-1H	One-side heating		1	1
OS-2H			2	1
OS-3H			3	1
TS-1H	Two-side heating		1	1
TS-2H			2	1
TS-3H			3	1

**Table 9 materials-14-01793-t009:** The temperature of inner concrete and rebar.

T/C No.	Concrete Cover of Exposed to Fire (mm)		Max Temperature (°C)					
			1 h (60 min)		2 h (120 min)		3 h (180 min)	
			OS	TS	OS	TS	OS	TS
**Slab**								
1	30		250	34	333	89	1061	124
2	60		134	30	192	68	783	114
3	80		110	28	122	55	674	92
4	140		49	27	100	37	579	46
5	190		29	27	77	31	326	31
6	240		26	27	42	28	671	28
7	340		26	27	25	27	549	27
**Wall**								
8	30		152	208	128	216	218	389
9	60		109	121	113	107	143	219
10	90		100	108	74	137	111	135
11	140		33	84	29	64	64	102
12	190		24	38	25	64	34	95
13	240		24	29	30	40	27	84
14	340		24	28	40	29	24	39
**Reinforced bar (Rebar)**								
15	Slab bottom	40	143	130	245	206	866	358
16	Slab top	420	26	28	24	28	45	35
17	Connection (Zone 1)		318	132	445	278	481	284
18	Connection (Zone 2)		26	132	26	278	26	284
19	Wall back (Zone 2)		26	208	445	216	33	389
20	Wall top	40	26	28	25	29	28	30
21		200	25	28	26	28	26	29
22		520	26	26	26	28	26	28

**Table 10 materials-14-01793-t010:** Reduction factor (k) of concrete yield strength (${f^{\prime }}_{c}$), rebar yield strength ($f_{y}$), and elastic modulus ($E_{c},\;E_{y}$) at elevated temperature (EN 1992-1-2 [17], ACI 216.1M-14 [18]).

**Heating Time (Hours)**	**Material**	Location	T/C No.	Concrete cover Of Exposed to Fire (mm)	Temperature (°C)			Reduction Factor						-
								${f^{\prime }}_{c},\;f_{y}$			$E_{c},\;E_{y}$			
					$T_{test}$	$T_{EC}$	$T_{ACI}$	$k_{f,Test}$	$k_{f,EC}$	$k_{f,ACI}$	$k_{E,Test}$	$k_{E,EC}$	$k_{E,ACI}$	
					A	B	C	D	E	F	G	H	I	Row
**1**	Concrete	Slab	1	30	250	400	420	0.76	0.62	0.60	0.64	0.39	0.36	1
	Rebar	Slab	15	40	143	300	320	1.00	0.98	0.98	1.00	1.00	1.00	2
		Connection	17	50	318	230	250	0.98	0.99	0.99	1.00	1.00	1.00	3
**2**	Concrete	Slab	1	30	333	560	600	0.65	0.43	0.38	0.48	0.14	0.08	4
	Rebar	Slab	15	40	245	460	480	0.99	0.97	0.97	1.00	1.00	1.00	5
		Connection	17	50	445	395	410	0.97	0.97	0.97	1.00	1.00	1.00	6
**3**	Concrete	Slab	1	30	1061	690	710	0.00	0.29	0.27	0.00	0.05	0.05	7
	Rebar	Slab	15	40	866	560	600	0.84	0.96	0.96	1.00	1.00	1.00	8
		Connection	17	50	481	500	520	0.97	0.97	0.97	1.00	1.00	1.00	9

**Table 11 materials-14-01793-t011:** Result of load–displacement.

Specimen		Load (kN, P)	$\mathrm{Load}\;\mathrm{Ratio}\;(P/P_{NS})$	Displacement (mm, δ)
**NS**		1,779.62 $\left(P_{NS}\right)$	1.00	18.11
**OS**	OS-1H	1757.56	0.99	12.40
	OS-2H	1926.21	1.08	13.65
	OS-3H	856.83	0.48	10.52
**TS**	TS-1H	1043.66	0.59	9.99
	TS-2H	901.98	0.51	15.39
	TS-3H	856.34	0.48	14.16

**Table 12 materials-14-01793-t012:** Result of moment–deflection angle.

Specimen		Moment (kN·m)	$\mathrm{Moment}\;\mathrm{Ratio}\;(M/M_{NS})$	Angle of Deflection (Radian, θ)		
				Left (A)	Right (B)	Difference (|A−B|)
**NS**		1276.88 $\left(M_{NS}\right)$	1.00	0.0284	0.0245	0.00390
**OS**	OS-1H	1261.05	0.99	0.0159	0.0155	0.00040
	OS-2H	1382.06	1.08	0.0201	0.0165	0.00357
	OS-3H	614.78	0.48	0.0189	0.0086	0.01029
**TS**	TS-1H	748.83	0.59	0.0043	0.0043	0
	TS-2H	647.31	0.51	0.0107	0.0104	0.00027
	TS-3H	614.56	0.48	0.0104	0.0089	0.00147

**Table 13 materials-14-01793-t013:** Load and moment according to the strength reduction factor for each standard (EN 1992-1-2 [17], ACI 216.1M-14 [18], and NIST 1681 [19]).

**Heating Time (Hours)**	Test				EN 1992-1-2				ACI 216.1M-14, NIST 1681				Max Difference	
	One-Side (OS)		Two-Side (TS)											
	Load (kN)	Moment (kN·m)	Load (kN)	Moment (kN·m)	Load (kN)		Moment (kN·m)		Load (kN)		Moment (kN·m)		Load (kN)	Moment (kN·m)
	$P_{Test,OS}$	$M_{Test,OS}$	$P_{Test,TS}$	$M_{Test,TS}$	$P_{EC}$		$M_{EC}$		$P_{ACI,NIST}$		$M_{ACI,NIST}$			
					$k_{f,Test}$	$k_{f,EC}$	$k_{f,Test}$	$k_{f,EC}$	$k_{f,Test}$	$k_{f,ACI}$	$k_{f,Test}$	$k_{f,ACI}$		
**0**	1779	1276	1779	1276	1589		1140		1020		719		759	557
**1**	1757	1261	1043	748	1584	1580	1137	1133	1020	1001	719	706	756	555
**2**	1926	1382	901	647	1584	1575	1137	1130	1011	992	713	699	1025	735
**3**	856	614	856	614	1515	1570	1087	1127	865	982	610	692	714	517

## Data Availability

Data are contained within the article.

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
