# Peer review of "Post Fire Residual Strength of the Wall-Slab Using Siliceous Concrete"

_materials, 2021, doi:10.3390/ma14071793_

Round 1
Reviewer 1 Report
The originality and the scientific value of the subject research are good.
The research area is the Post Fire Residual Strength of the Wall-Slab.
The research area addressed is current.
The solution of research and experiments is logical.
The research is documented in detail and presented on 33 pages of the manuscript.
Some information is already known.
Horewer, manuscript also brings valuable new information.
Please add the standard deviation, number of samples, and CoV for experiments/mechanical properties if possible.
Were other mechanical properties of concrete tested - tensile strength?
The part of the Discussion must be presented separately. Authors must evaluate the results achieved and state the results in the context of current research.
A comprehensive solution to the problem would also require the use of advanced numerical modeling and nonlinear analysis for said performance-based-design . Also considering that the mechanical properties of concrete have a stochastic character.
Authors must also add information - references to testing and numerical modelling of reinforced concrete structures in the part of the Introduction.
Vecchio, F.J.; Shim, W. Experimental and analytical reexamination of classic concrete beam tests.Journal of Structural Engineering 2004, 130 (3), pp. 460-469.
Sucharda, O. ; Konecny, P.. Recommendation for the modelling of 3D non-linear analysis of RC beam tests. Comput. Concr. 2018, 21, 11–20.
Overall, it is possible to evaluate the experimental program positively, but the authors must improve the presentation of the research and the informative value of the manuscript.
It is not possible to publish an article in its current form.
The manuscript must be revised before publication.
Author Response
Dear Reviewer 1,
Thank you for taking the time to review the manuscript.
I checked the comments given to the author.
Content revised in the manuscript due to the comments of reviewers is indicated in red letters.

Reviewer 2 Report
the current study investigates the reinforced concrete performance made from reinforced concrete wall reinforced concrete slab connection to determine the structural residual and fire performance. The authors carry out fire scenarios acting on one and two sides to assess the strength of WSC.
Please consider reviewing the abstract and highlight the novelty, major findings and conclusions.
Abstract needs tidying and rephrasing for example check line 16/17 the sentence can be shortened and improved
In the introduction, what is the research gap did you find from the previous researchers in your field? Mention it properly. It will improve the strength of the article. Mention what past studies did, what were their main findings and how does your work bring new knowledge to the field.
Figure 1 and everything related to is more suited for appendix section, please create one and move the figure
Line 155 “s, it was confirmed that the concrete exposed to high temperature had lower 155 strength after cooling” what effect does this have on the performance of the concrete slab/wall
Section 3.2 Wall-Slab Connection Specimen Fabrication Plan what are the limitation in the experimental setup in this study in comparison to real life event in a building under fire
What is the maximum operating temperature of the K-type thermocouples?
Line 391-392 was this observed in past studies similar to your work? If yes then discuss this further and compare your work with the results from past study.
Line 560 “the strength was significantly reduced” please describe in terms of percentage or numbers instead of saying significantly.
This is very long study which can be broken into two different studies, the readers might lose interest in the manuscript due its length.
What is the research gap did you find from the previous researchers in your field in each section you discussed and analyse? Mention it properly.
The results are well discussed and there is some attempt of going the extra mile to critically discuss and analyse the data with past studies and standards but there are some other sections where the results are merely described and is limited to comparing the experimental observation or reporting it. The authors are encouraged to include more discussion section and critically discuss the observations from this investigation with existing literature.
Author Response
Dear Reviewer 2,
Thank you for taking the time to review the manuscript.
I checked the comments given to the author.
Content revised in the manuscript due to the comments of reviewers is indicated in red letters.

Round 2
Reviewer 1 Report
Thanks for the comments and manuscript edits.
The presentation of the research and results is also at a good level.
The manuscript can be accepted for publication.
Reviewer 2 Report
all questions answered